# NgIohTuned: a Highly Reproducible Black-box Optimization Wizard

## Abstract

Inspired by the observations in neuro-control and various reproducibility issues in black-box optimization in the machine learning community, we revisit black-box optimization with rigorous benchmarking in mind. We (i) compare real-world (RW) benchmarks with artificial ones, emphasizing the success of Differential Evolution, Particle Swarm Optimization, and bet-and-run in the former case; (ii) introduce new artificial benchmarks, dubbed *multi-scale benchmarks*, with a focus on scaling issues (where scale refers to the unknown distance between the optimum and the origin), akin to real-world benchmarks such as those arising in neural reinforcement learning; (iii) demonstrate the performance of quasi-opposite sampling and of mathematical programming methods (Cobyla and direct search) on multi-scale continuous benchmarks; (iv) showcase the robustness and performance of algorithms focusing on a carefully chosen decreasing schedule of the mutation rate on discrete benchmarks; (v) design novel continuous black-box optimization strategies combining optimization algorithms with good scaling properties in the first phase, robust optimization techniques in the intermediate phase, and methods with fast convergence in the final optimization phase. Our methods are included in a public optimization wizard, available in two versions: NgIoh4 (which does not leverage information about the type of variables) and NgIohTuned (leveraging all conclusions of the present paper, including choosing algorithms thanks to high-level information on the real-world nature of a problem and/or its neuro-control nature and applying recent direct-search methods). They are integrated into a platform with complete reproducibility on a large benchmarking suite.

## 1 Introduction

Black-box optimization is the optimization of functions on discrete or continuous domains without any gradient or white-box information. Inspired by the Dagstuhl seminar 23251 *Challenges in Benchmarking Optimization Heuristics* (July 2023), we develop additional benchmarks within the black-box optimization platform Nevergrad (Rapin & Teytaud, 2024) (including problems from reinforcement learning, tuning for machine learning, planning, etc.) and improve algorithms accordingly. The contributions of this paper are twofold.

**First, benchmarks investigation.** After formalizing black-box optimization (BBO) benchmarks (Section 4.1), Section 4.2 discusses the necessary diversity of benchmarking suites. Then, following (Meunier et al., 2022), we observe that one can significantly alter the conclusions of a benchmark by changing the distribution of the optima, in particular by scaling its variables (e.g., by placing them closer to zero or closer to the boundary of the domain). The distribution of the optima, typically induced by the random shifts used in benchmarking platforms, has a major impact on the experimental results. Section 4.2 illustrates this issue using multi-scale benchmarks with results described in Section 5.1, thereby avoiding comparisons where algorithms overfit to a specific scale. We also analyze key insights brought from real-world benchmarks, as discussed in e.g., (Ungredda et al., 2022; Dagstuhl participants, 2023). Section 5.2 and Appendix G present results in real-world cases within Nevergrad.

**Secondly, algorithmic design exploration.** Section 2.2 reviews algorithms that perform well on a variety of multiscale benchmarks (discrete/continuous domains, synthetic/real-world scenarios). These approaches are integrated into a state-of-the-art optimization wizard (see Section 2) for black-box optimization that improves over state-of-the-art algorithms on average on numerous benchmarks (results in Section 5 and later): NgIoh4, which incorporates our improvements regarding multi-scale problems into the existing NGOpt wizard (as detailed in Section 3), and NgIohTuned, which incorporates further algorithmic enhancements (specifications in Section 3.1).

## 2 Background & Related Works

Black-box optimization (BBO) tasks arise in various components of AI, such as reinforcement learning, hyperparameter tuning, and planning. The *scale* of a black-box optimization problem, which corresponds to the distance between its optimal solution(s) and the origin, has a major impact on algorithmic performance.

### 2.1 Black-box optimization wizards

**Scaling and mutation rates.** Rahnamayan et al. (2007) focus on initializing population-based methods for robustness to the scale in the continuous context. In the discrete case, Doerr et al. (2019); Einarsson et al. (2019); Doerr et al. (2017b); Dang & Lehre (2016) are entirely based on scheduling the scale of mutations. Methods focusing on a fixed schedule are particularly robust in the discrete setting, especially when compared to adaptive methods (Kruisselbrink et al., 2011a), which have excellent results in some cases.

**Black-box based optimization algorithms.** In terms of continuous BBO methods, Differential Evolution (DE) (Storn & Price, 1997) and Particle Swarm Optimization (PSO) (Kennedy & Eberhart, 1995) are well-known. Compared to CMA (Hansen & Ostermeier, 2003), the method focuses on rapidly approximating the appropriate scale and is well-suited for high-dimensional settings. In contrast, CMA is primarily robust against conditioning and rotation challenges. Bayesian methods (Jones et al., 1998) and machine learning-based methods are another branch of the state-of-the-art: among them, SMAC3 (Lindauer et al., 2022) and HyperOpt (Bergstra et al., 2015) perform particularly well. Cobyla (Powell, 1994) comes from the mathematical programming community and frequently performs well in low-budget cases (Raponi et al., 2023). Sequential Quadratic Programming (SQP) is another well-known approach with an excellent local convergence rate (Nocedal & Wright, 2006). Several direct-search methods are already included in Nevergrad, and we add a recent one from (Roberts & Royer, 2023). A summary of the other standard optimization techniques considered in this paper is given in Appendix I.

**Black-box optimization wizards.** Recently, optimization *wizards* (inspired by algorithms from other areas such as (Xu et al., 2008)) have become common. These tools combine various base algorithms, designed to be immediately effective on a wide range of benchmarks without tuning, regardless of noise, parallelism, computational budget, variable types, or number of objectives. Wizards typically rely on a diverse set of static portfolio strategies (Liu et al., 2020; Meunier et al., 2022) and employ bet-and-run techniques (Weise et al., 2019) We note that the best-performing method in the BBO challenge (AX-team, 2021) is a wizard termed Squirrel (Awad et al., 2020) combining, among others, DE and SMAC3.

**Black-box optimization platforms** In terms of platforms, many libraries exist (e.g., (Johnson, 1994; FacebookResearch, 2020; Virtanen et al., 2020)). Nevergrad (Rapin & Teytaud, 2024) imports these libraries and others. Concerning benchmarks/applications, the BBO Challenge (AX-team, 2021) (close to real-world, with best performance obtained by a wizard including differential evolution in Awad et al. (2020)), Keras (Chollet et al., 2015), scikit-learn (Pedregosa et al., 2011), COCO/BBOB (Hansen et al., 2009a) (artificial, best performance by CMA variants (Hansen & Ostermeier, 2003)), LSGO (Li et al., 2013), IOH (Doerr et al., 2018), OpenAI Gym (Brockman et al., 2016) are well known. The benchmarks in the present paper include them or some of their variants and many others, and, with our present work, including quasi-opposite forms of DE, SQOPSO (Zhang et al., 2009), NgIoh wizards mentioned above, Carola algorithms (see Section 2.2 for these algorithms) and new benchmarks including multi-scale benchmarks (Section 4.2).

## 2.2 Algorithms for multi-scale benchmarks

We highlight here only a few selected algorithms. A brief description of each algorithm used can be found in Appendix I.

**Opposite and quasi-opposite sampling.** Rahnamayan et al. (2007) propose to initialize the population in DE as follows: (i) randomly draw half the population independently identically distributed (as usual) and (ii) for each point $x$ in this half population, also add $-x$ (opposite sampling) or $-r \times x$ (quasi-opposite sampling, where $r$ is chosen i.i.d. uniformly at random in the interval $[0, 1]$). A key advantage of the quasi-opposite method is that the resulting population includes points with many different norms, which is beneficial for multi-scale settings. We use quasi-opposite sampling in DE and PSO, with variants termed QODE, QNDE, SPQODE, LQODE, SODE, QOTPDE, QOPSO, SQOPSO, fully described in Appendix C. SQOPSODCMA is a chaining: it contains SQOPSO, followed by diagonal CMA for local convergence. We observe good results, overall, for SQOPSO and various quasi-opposite tools (Section 5.3), in particular in the real-world context (Section 5.2), including neuro-control (Section 5.2 and Appendix G.5).

**Direct-search methods.** New tools from the direct-search community have been added into the direct-search part of Nevergrad, namely the Direct-search (DS) methods from (Roberts & Royer, 2023). These methods are based on iteratively moving (polling) along fixed or varying directions. The moves are controlled by means of a stepsize sequence, and are accepted if they reduce the function by any amount (simple decrease) or a prescribed amount (sufficient decrease). The included methods are:

- DSbase: Method based on sufficient decrease and adaptive step size. At every iteration, polls along all coordinate directions and their negatives.

- DSproba: Method based on sufficient decrease and adaptive step size. At every iteration, polls two opposite random directions uniformly distributed on the unit sphere.

- DSsubspace: Method based on sufficient decrease and adaptive stepsize. At every iteration, polls opposite Gaussian directions generated in a random subspace (default: 1D subspace).

- DS3p: Method based on simple decrease and predefined, decreasing step size sequence ($\alpha_0/(k+1)$ at iteration $k$). At every iteration, the method polls opposite Gaussian directions.

DSproba and DSsubspace have analogies and differences with the $(1 + 1)$ evolution strategy. DSproba is randomized, whereas DSsubspace is not. Both accept an iterate only if it satisfies a sufficient decrease condition (essentially, reducing the function value by an amount proportional to the square of the stepsize). Opposite directions are used in DSproba/DSsubspace, which is crucial for the theoretical guarantees to hold.

We here test them on our multi-scale benchmarks (MS-BBOB and ZP-MS-BBOB) for which several known algorithms have difficulties if they do not know the scale a priori. Results are shown in Figure 1.

**Continuous multi-scale benchmarks.** Cobyla is good when the scale of the optimum is unknown, as shown by later results and by (Dufossé & Atamna, 2022; Raponi et al., 2023), and we have seen that quasi-opposite sampling is designed for helping DE in that same context. Another solution for guessing the scaling of the optimum is to assume that the scaling of the optimum $x$ for different variables might be similar, i.e., $\log |x_i| \simeq \log |x_j|$ for $i \neq j$. Inspired by this observation, we propose RotatedTwoPointsDE, a variant of DE using a 2-point crossover (Holland, 1975), with the possibility of moving the cut part to other variables. Thus, more precisely, DE typically mixes the $i^{th}$ variable of an individual and the $i^{th}$ variable of another individual and the child gets the result at the $i^{th}$ position (Algorithm 3). This happens for several indices $i$, but the $i^{th}$ variable has no impact on the $j^{th}$ variable if $j \neq i$. TwoPointsDE uses the two-points crossover, which has a similar property: the difference with the classical DE is that the variables impacted by the crossover are in a segment of consecutive variables rather than randomly distributed in the list of variables. Both DE and TwoPointsDE find scales by working somehow separately on variables. RotatedTwoPointsDE can move this segment of consecutive variables, and therefore it might combine the $i^{th}$ variable of an individual and the $i^{th}$ variable of another individual and the child gets the result at the $j^{th}$ position where $j = i + k$ (modulo

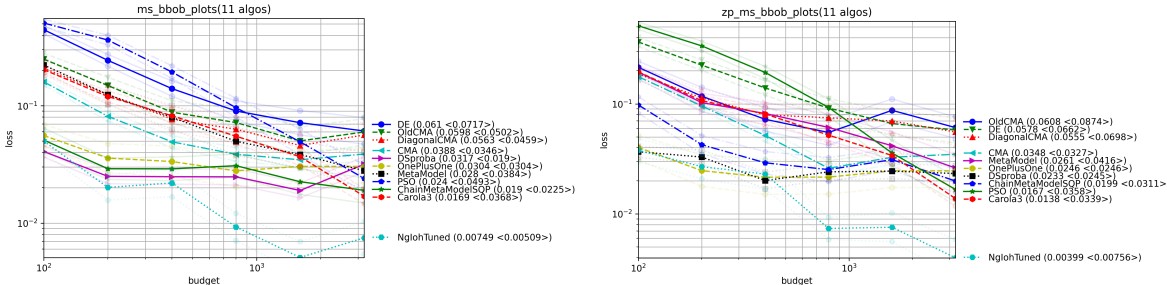

Figure 1: **NgIohTuned and DSproba tested on continuous multi-scale benchmarks** (average normalized loss as in Section 4.3). DSproba (Roberts & Royer, 2023) performs well on MS-BBOB and ZP-MS-BBOB compared to all base algorithms. Carola3 is good, though weaker than DSproba for low budgets: compared to DSproba, it can be used also in the parallel case. NgIohTuned, which is a wizard using all improvements described in the present paper (DSproba, Carola3, and leveraging high-level information on the nature of the problem for switching to quasi-opposite initializations and bet-and-run in neuro-control and other real-world problems), outperforms all methods on all budgets, including the previous wizard NGOpt.

---

**Algorithm 1** Three variants of Carola. MetaModel refers to the MetaModel implementation in (Rapin & Teytaud, 2024), based on quadratic approximations built on the best points so far.

---

Carola1:

**Require:** Budget $b$
  Apply Cobyla with budget $b/2$.
  Apply CMA with Meta-Model with budget $b/2$ and initial point the best point so far.

Carola2:

**Require:** Budget $b$
  ***Fast approximation:*** apply Cobyla with budget $b/3$.
  ***Robust local search:*** Apply CMA with MetaModel with budget $b/3$ and initial point the best point so far.
  ***Fast local search:*** Apply SQP with initial point the best point so far and budget $b/3$.

Carola3:

**Require:** Budget $b$, number $w$ of workers
  Apply $w$ copies of Carola2 in parallel, with budget $b/w$

---

the number of variables) for some $k \neq 0$. The assumption behind RotatedTwoPointsDE is that the scale is not totally different, at least in terms of order of magnitude, for different variables: we can carry variables from a position to another. Another variant, GeneticDE, uses RotatedTwoPointsDE during 200 candidate generations for finding the correct scale, before switching to TwoPointsDE.

**The scaling of mutations in the context of discrete optimization.** In discrete optimization, the good old $1/d$ mutation consists in randomly mutating each variable with probability $1/d$ in dimension $d$. Typically, a single variable is mutated; it rarely includes more than two variables. Some algorithms, in particular after the good results described in (Dang & Lehre, 2016), use a fixed random distribution of mutation rates. The adaptation of FastGA (Doerr et al., 2017b) in Nevergrad consists in randomly drawing a probability $p$ (instead of using $p = 1/d$) in $[0, \frac{1}{2}]$ (in $[0, 1]$, if the arity is greater than two). DiscreteLenglerOnePlusOne, inspired by (Einarsson et al., 2019), consists in using a schedule. In this case, the probability $p$ decreases during the optimization run.

## 3 NgIoh

From observations on IOH (Doerr et al., 2018), we propose some new principles for the design of BBO wizards. NGOpt (Nevergrad Optimizer) is the current wizard of Nevergrad, and NGOptRW is another wizard, specialized on real-world benchmarks, doing a bet-and-run between DE, PSO and NGOpt during

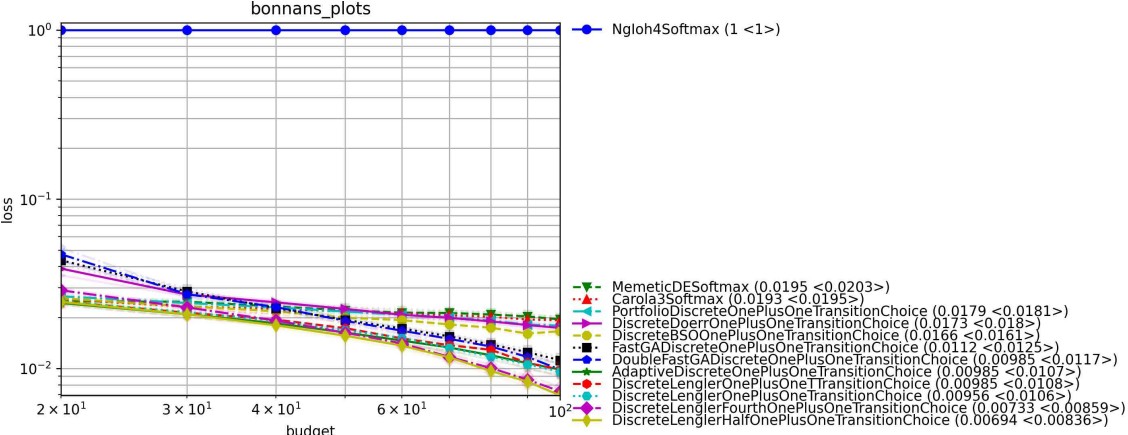

Figure 2: Various methods (86 algorithms run; only the best 12 ones are presented, and the single worst) on the Bonnans (discrete) function (average normalized loss as in Section 4.3). The softmax representation (i.e., converting the problem to a continuous one, as optionally proposed in Nevergrad) performs poorly here compared to the standard discrete representation (termed TransitionChoice in Nevergrad). The Discrete-LenglerOnePlusOne method (and its variant with modified parameters, with similar names) performs well on Bonnans functions (Bonnans et al., 2023).

33% of the budget before switching to the best of them (unless the benchmark is noisy or discrete, in which case it simply uses NGOpt).

First, whereas it is classical (e.g., Memetic algorithms (Moscato, 1989)) to run evolution strategies first and local methods afterwards (as Nevergrad's NGOpt frequently does), we observe that Cobyla is excellent for low budget and multi-scale problems and looks like a natural candidate for a warmup stage before other optimization tools. Therefore, we proposed Carola (Cost-effective Asymptotic Randomized Optimization with Limited Access), a method running Cobyla first and then other methods, as presented in Algorithm 1.

Second, our insights are gathered in a new BBO wizard, which we dub NgIoh. Ng stands for Nevergrad (Rapin & Teytaud, 2024), while Ioh stands for IOH (Doerr et al., 2018). NgIoh differs from NgOpt by applying Carola methods instead of CMA + metamodels in sequential cases in which NGOpt was using such a variant of CMA. NgIoh4 is basically the same as NGOpt (Nevergrad's wizard), except that it switches to Carola2 depending on some rules (see Algorithm 2). The constants in the rules were chosen based on the observations described in (Doerr et al., 2018). The different variants (performing similarly) are available in (Rapin & Teytaud, 2024). NgIoh4 performs slightly better overall, so we keep it as a reference in our experiments. More details about NgIoh variants can be found in Appendix I.

---

**Algorithm 2** NgIoh4, combining NGOpt and ideas extracted from results in IOH (Doerr et al., 2018).

**Require:** Budget $b$, dimension $d$, domain $D$, number $w$ of workers.
  **if** $w = 1$ and $D$ is continuous and the problem is deterministic and ($d < 100$ and $20d \leq b \leq 1000d$) or ($d < 50$ and $b < 1000d$). **then**
    Apply Carola2.
  **else**
    Apply NGOpt.
  **end if**

---

**Ablation for Carola4.** Carola3 is an adaptation of Carola2 for the parallel case, so let us focus on the comparison between Carola1, Carola2, and algorithms on which they are based, namely Cobyla, CMA and MetaModel. Essentially, our results in Figure 3 validate Carola2: we refer to Appendix A for more variants and their results.

**Ablation for NgIoh variants.** We tested NgIoh4 and variants (Wiz, NgIoh2, NgIoh3, NgIoh5, NgIoh6, NgIoh and others): they are defined in Appendix I and compared in Figure 3 and Appendix F: basically, we validate NgIoh4.

### 3.1 NgIohTuned

NgIoh4 is important for validating Carola and its warmup by Cobyla, with all other parts of NgOpt being equal. However, it makes sense, once this validation is made, to switch to an optimized wizard. So, we also define the NgIohTuned wizard (more details in Appendix H). NgIohTuned is similar to NgIoh4, but leverages ideas derived from external benchmarks (Appendix G) and recent papers:

- DSproba is a promising direct-search method for low budgets, more recent than the direct-search methods that were available in Nevergrad when NGOpt was built. Given results in Figure 1, NgIohTuned uses DSproba instead of CMA in sequential continuous cases when the budget is lower than the dimension.

- Based on the results described in Figure 24 (Appendix H), we observe that for large budgets, we might afford large population sizes which improve the robustness: so NgIohTuned uses VLPCMA (i.e., CMA with population size multiplied by 100) instead of CMA when the budget is greater than 2000 times the dimension.

- The parameters of the chaining in Carola are tuned: 10% for Cobyla, 80% for the CMA with MetaModel, 10% for the final convergence with SQP, instead of 1/3 for each.

- Given the results on real-world benchmarks in Appendix G, NgIohTuned leverages additional information provided by the user (if any) for switching to different algorithms: it switches to SQOPSO (for deterministic neural control) and to NGOptRW (for other deterministic continuous real-world problems).

## 4 Experimental Setup

Motivated by recent warnings such as Kapoor & Narayanan (2023); Li & Talwalkar (2019), we first take a moment in Section 4.1 to reflect on reproducibility. Then, we present the selected benchmark suites (Section 4.2) and the algorithms.

### 4.1 Reproducibility

Reproducible benchmarking is essential for scientific development, yet the machine learning community often faces poor reproducibility practice, especially when dealing with black-box optimization (see discussion in (Swan et al., 2022; Bäck et al., 2023)). A discussion of poor reproducibility in deep learning-assisted optimization is available in (Markov, 2023); see also (Liu et al., 2021; Li & Talwalkar, 2019; Pham et al., 2018; Real et al., 2019). These works show how some simple methods might, in spite of promising claims, outperform black-box optimization methods backed by intensive GPU-based machine learning. In addition, Kapoor & Narayanan (2023); Haibe-Kains et al. (2020) mention various examples of results in the machine learning community that are difficult to reproduce. Meunier et al. (2022) mentions misleading results when initialization issues alter baselines in some papers.

We propose a code fully available in open access. A PDF with all experimental results is available at `tinyurl.com/dagstuhloid`. Though our focus is on the *ability to rerun everything*, the entire data is available at `tinyurl.com/bigdagstuhloid`[1]. As these URLs are automatically updated, they might differ thanks to additional work by contributors and re-runs, so upon acceptance of this submission, a "frozen" version of code and data will be store in a permanent storage facilities such as Zenodo. Similarly, packages version (e.g., PyPI) can have an impact on the results, and dependency upon a specific version of a given package is detrimental to reproducibility. Thus, following practice in Nevergrad, our code requires a minimum version

---

[1]Warning: $> 300$MB, representing data from more than 20 million runs.

number for each package rather than a fixed version number. Further details about reproducibility of our results are mentioned in Appendix B.

## 4.2 Benchmark Suites (a.k.a. Problem Collections)

We aim to develop a diverse set of benchmark suites that encompass a wide range of problem settings encountered in practice. This includes diversity regarding budget, performance measure, distribution of the optima, among others. Table 1 summarizes the diversity of our benchmarks and their parameters. For each benchmark suite, the detailed setup is described on the GitHub page at `tinyurl.com/2p8xcdrb`.

**Budgets and diversity.** Ungredda et al. (2022); Dagstuhl participants (2023) showed that cases with budget with more than 100 times the dimension (which is frequent in artificial benchmarks) might be the exception rather than the norm. In real-world applications, we may even face settings in which the total number of function evaluations may not exceed a fraction of the dimension. We therefore consider a large variety of different scalings of the budget, including cases with budget far lower than the dimension. More generally, Appendix D shows the diversity of the benchmarks used in our results.

**Scaling and distribution of optima in continuous domains: multi-scale problems.** Although the key issue of the optimum at zero is solved by randomizing its position (Kůdela, 2022; 2023), the way the optimum is randomly drawn brings new issues. Throughout the discussion, we assume that the center of the domain is zero. This is not the case in all benchmarks and assumed without loss of generality: this is just a simplification for shortening equations, so that we can use $-x$ for symmetries instead of $2c-x$ with $c$ being the center, and $||x||$ instead of $||x - c||$. We observe that scaling is an important issue in benchmarks. Typically, in real-world scenarios, we do not know in advance the norm of the optimum (Kumar, 2017): in particular, initializing different algorithms with different scales (and not the best one for baselines) makes conclusions unreliable, as pointed out in Meunier et al. (2022). Assuming that the optimum has all coordinates randomly independently drawn with center zero implies that the squared norm of the optimum is, nearly always, close to the sum of variances: this is the case in many artificial benchmarks. Consequently, it reduces the generality of results: conclusions drawn on such benchmarks are valid essentially on problems for which there is a nearly constant norm of the optimum. To address these issues and drawing inspiration from various real-world benchmarks (Cotton, 2020a;b; Raponi et al., 2023), we propose the following new artificial benchmarks:

**Different distributions of the optimum: MS-BBOB.** MS-BBOB is quite similar to BBOB (Hansen et al., 2009b) or YABBOB (Rapin & Teytaud, 2024). However, MS-BBOB (multi-scale BBO benchmark), has different scales for the distribution of the optima. This is done by introducing a scaling factor $\tau$ which varies in $\{0.01, 0.1, 1.0, 10.0\}$. This scaling factor is applied to the random drawing of the optima. For example, in some benchmarks, Nevergrad uses a normal random variable as a shift for choosing the optimum. Thus, this random variable is multiplied by $\tau$.

**Zero-penalization: ZP-MS-BBOB.** Many benchmarks, including our benchmarks in MS-BBOB, are symmetrical w.r.t. zero. The optimum might be translated, but that translation has zero mean. This special role of the center might imply that the neighborhood of zero provides too much information. Actually, many real-world problems have misleading values close to zero, in particular in control or neuro-control (e.g., for neuro-control the control is just zero if all weights in a layer are zero). Therefore, we consider zero-penalized problems: this means we add a penalty close to the optimum (which changes the actual optimum), so that the shape of the objective function close to zero cannot tell the algorithm where is the optimum. This variant is called ZP-MS-BBOB (zero-penalized MS-BBOB).

**Real-world benchmarks.** We use the suffix "(RW)" to denote real-world benchmark problem. Note that the definition of "real-world" is not so simple. All experiments are entirely in silico, and in some cases the model has been simplified. RW thus means that we consider the benchmark as sufficiently real-world for being tagged as such. Our experiments include neuro-control with OpenAI Gym (Brockman et al., 2016), policy optimization with Aquacrop (Raes et al., 2009), PCSE (de Wit, 2021), and hyperparameter tuning with Keras (Chollet et al., 2015) and scikit-learn (Pedregosa et al., 2011). Note that an additional real-world

benchmarking is performed in Appendix G, for checking the validity of our conclusions on completely distinct problems outside Nevergrad.

## 4.3 Performance criteria

For each benchmark, we consider two figures. First, the **Normalized simple regret figure.** A convergence curve, with the budget on the x-axis and the average (over all budgets) normalized (linearly, to $[0, 1]$) loss. Note that some benchmarks do not have the same functions for the different values of the budget. Therefore, we might have a rugged curve, not monotonous. This is even more the case for wizards such as NGOpt or NGOptRW, which make decisions based on the budget: they might make a bad choice for some values of the budget, leading to irregular curves. Curves are named with (i) the algorithm name (ii) the average normalized loss for the maximum budget of the experiment, and (iii) the average normalized loss for the penultimate budget of the experiment. The point of the score for the penultimate budget is to check the stability between the penultimate and the last budget.

Second, in appendix only, the **Frequency of winning figure.** A heatmap, showing the frequency $f_{m,m'}$ at which a method $m$ (row) outperforms on average (over the different problems and replicas) another method $m'$(column). Frequencies are computed over all instances and all budgets. Methods are then ordered by the average $score_m$ of these frequencies $f_{m,m'}$ over all other methods $m'$. The columns show the names of the methods, appended with the number of settings they were able to tackle (for example, some methods have no parallel version and therefore do not run on all settings).

The complete archive (see links in Appendix B) shows many competence maps. Given the hyperparameters of a benchmark (e.g., dimension, budget, level of noise, among others), the competence maps in the full archive show, for a given pair of hyperparameter values, which algorithms perform the best on average.

## 5 Experimental Results

We believe that Section 4.2 points out an important and generic issue for improving BBO benchmarks: so, Section 5.1 presents multi-scale continuous benchmarks, added inside Nevergrad. Section 5.2 presents the results on real-world benchmarks: they illustrate the gaps between real-world and artificial benchmarks. Then, Section 5.3 validates our wizards on all benchmarks.

The appendix contains additional figures for a more extensive view of our results, and detailed data can be found in the automatically generated `tinyurl.com/dagstuhloid`. All our code is merged in Nevergrad (Rapin & Teytaud, 2024), and `tinyurl.com/2p8xcdrb` contains a description of all benchmarks integrated in Nevergrad (previous benchmarks and ours).

## 5.1 Multi-scale BBO benchmarks: dealing with the scaling issues, and validating Carola and NgIoh

In the case of continuous optimization, we present new benchmarks, adapted from YABBOB (Rapin & Teytaud, 2024) using comments from Section 4.2. While CMA variants dominate in BBOB (Hansen et al., 2009a) (small scale, large budget, focus on frequency of solving with a given precision) and DE variants dominate in LSGO (Li et al., 2013) (larger scale, groups of variables), we propose a benchmark close to BBOB or YABBOB, but with a specific effort to not make the scale of the norm of the optimum to be known in advance (as detailed in Section 4.2). The principle of our proposed and open-sourced multi-scale BBOB (MS-BBOB) benchmark is that it contains four different scales (0.01, 0.1, 1.0, 10.0), and the algorithms are not informed of which scale is used for each instance. We also apply the zero-penalization, as discussed in Section 4.2. The resulting benchmark is termed ZP-MS-BBOB.

Experiments are presented in Figure 3. The best methods are all based on Carola (Section 3), on NgIoh (which introduces Carola inside NGOpt), or on quasi-opposite sampling. We conclude that the Carola method (using Cobyla first) and quasi-opposite samplings are both good for adapting a method to the right scaling of the variables. Carola2 is the best of the Carola variants for most budgets by a small margin (Figure 3, and Figure 18 in appendix). Quasi-opposite PSO variants perform well in these benchmarks, whereas it does not always perform well on non-multi-scale benchmarks: this coincides with the results

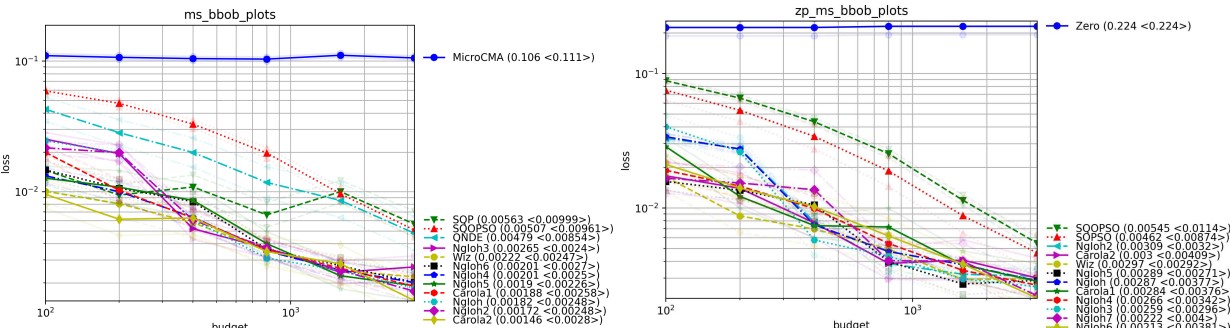

Figure 3: **Comparing Carola variants and NgIoh variants (except NgIohTuned, later) on multi-scale benchmarks** (average normalized loss as in Section 4.3). The best performing methods for MS-BBOB (left) and for ZP-MS-BBOB (right) for normalized regret. Both: we include the 12 best methods and (for reference) the worst: NGOpt and CMA are not in the 12 best, e.g., on the right NGOpt is ranked 15th and CMA is ranked 49th (out of 58 methods: more details in Figure 18, and more complete results including NgIohTuned in Figure 17, and an ablation in Figure 6). The good performances of Carola and NgIoh variants (including Wiz, also based on Carola2) are visible in both.

described in (Raponi et al., 2023), and it suggests that PSO is less robust to rotation but good for multi-scale problems. Appendix H.1 presents additional comparative results quantifying how much our tools dedicated to multi-scale problems work better than previous tools on multi-scale benchmarks.

**Regression: validating NgIoh variants in non multi-scale benchmark.** Figure 4 also shows that we do not deteriorate the performance too much on YABBOB (which does not have a multi-scaling) and Figure 5 shows that we improve results on the complete family of benchmarks (in particular with NgIohTuned, right of Figure 5). For additional regression testing on non multi-scale benchmarks:

- Figures 7 and 8 in the appendix present outcomes for various YABBOB variants in Nevergrad.

- Appendix H.3 confirms our results on COCO/BBOB benchmarks.

- Moreover, Section 5.3 demonstrates that, among the extensive family of benchmarks in Nevergrad, NgIoh4 outperforms NGOpt.

These results indicate that the results are not deteriorated when compared to NGOpt.

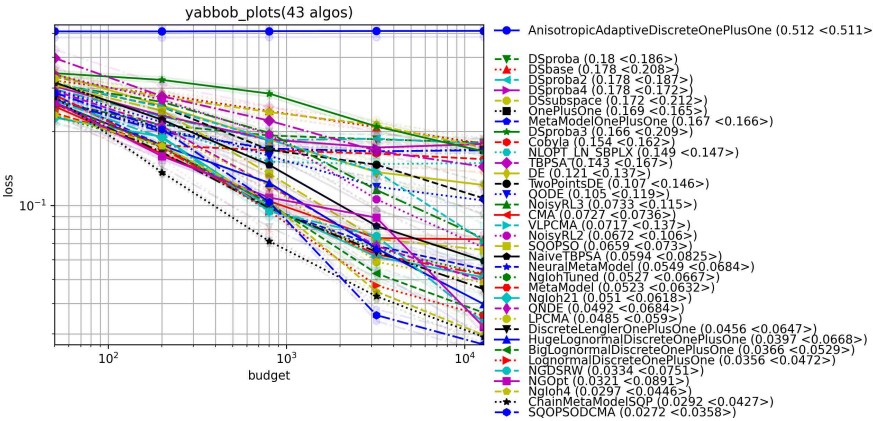

Figure 4: YABBOB, original benchmark (average normalized loss as in Section 4.3): our codes are less impressive on this initial benchmark not equipped with multi-scaling, but they still perform well. In Figure 5 we observe that on average on many benchmarks NgIoh4 outperforms NGOpt (whereas it just differs by integrating Cobyla as a warmup in Carola methods). We also observe in Figure 5 that NgIohTuned (which, compared to NgIoh4 which only leverages the multi-scale part of the present work, leverages all our conclusions, including taking into account the real-world nature and/or neuro-control nature of problems) outperforms all other methods by a bigger margin.

## 5.2 Real-world benchmarking in Nevergrad: successes of DE, PSO, bet-and-run, quasi-opposite sampling

We present in Figure 5 (left) the number of times each algorithm was ranked best among the list of real-world benchmarks in Nevergrad. Other real-world tasks (external to the Nevergrad benchmarks) are available in Appendix G and confirm our conclusions on these independent benchmarks.

For the real-world benchmarks of Nevergrad, we note that PSO and DE variants (in particular those with quasi-opposite sampling) perform better than in artificial benchmarks. We also note that NGOptRW, designed by adapting NGOpt for real-world instances by bet-and-run (Weise et al., 2019) of PSO/DE/NGOpt, performs very well. NGOptRW runs these three algorithms for one third of the budget, and keep the best of them for the rest. It vastly outperforms NGOpt and all others on RW benchmarks, though Carola3 (the possibly parallel adaptation of Carola2) is not bad.

## 5.3 Statistics over all Nevergrad benchmarks: validating NgIoh and NgIohTuned

**Comparison without knowing the type of problems and the type of variables** In Figure 5 (middle), we consider the number of times each method was ranked first when we add in Nevergrad solely our methods designed for multi-scale benchmarks, namely Carola and NgIoh (excluding NgIohTuned). NgIoh4 performs best with 10 times the first position, with a clear gap specifically for problems such that the scale of the optimum cannot be known in advance, i.e., multi-scale benchmarks (as detailed in Section 5.1). As an ablation, we build several variants, which perform similarly: NgIoh4 performs slightly better than or equivalently to other NgIoh variants (NgIoh5, NgIoh6, Wiz) and outperforms NGOpt: there is a domination of NgIoh variants. We note than due to many artificial benchmarks, NGOpt is here better than NGOptRW. NgIoh4 is also the best for the number of times it is ranked in the top 2 and for the number of times it is ranked in the top 3: the full details are reported in Appendix E. NgIoh4 performs even better if we remove the variants Wiz, NgIoh6, and NgIoh5 (documented in the codebase (Rapin & Teytaud, 2024)) from the statistics because they are quite similar.

**When we allow algorithms to use all available information.** When we allow algorithms to use the information "is real-world" or "is neuro-control" (Figure 5, right), NgIohTuned outperforms all other

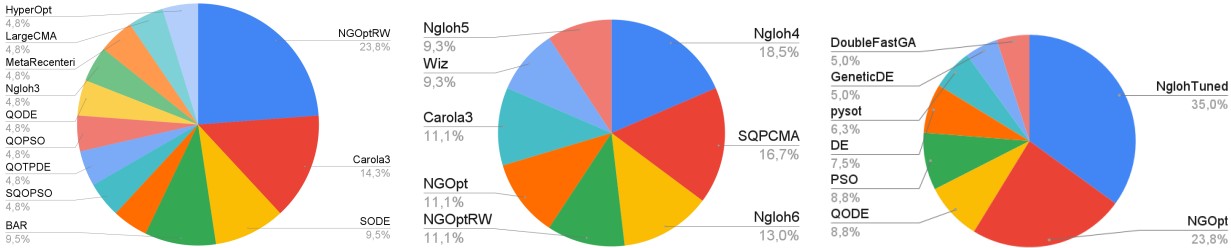

Figure 5: Statistics aggregated over Nevergrad benchmarks. **Left, real-world:** number of times each algorithm is ranked best, on the real-world part of the Nevergrad benchmarks (before adding NgIohTuned): NGOptRW clearly dominates, quasi-opposite tools are good, and NGOpt disappears: this validates the choices made by NgIohTuned in real-world cases. **Middle, all benchmarks, no information on the type of problem:** number of times each algorithm is ranked best (limited to algorithms ranked 1st at least 5 times), in the complete Nevergrad benchmarks, without including algorithms (such as NgIohTuned) using high-level information: NgIoh4 performs best. **Right, all benchmarks, full information:** number of times each algorithm is ranked best in the complete Nevergrad benchmarks when we remove Carola algorithms and NgIoh4 and just compare our main tool NgIohTuned (allowed to use information on the type of problems) to the previous methods: NgIohTuned dominates.

methods. This shows that such prior knowledge is relevant for a wizard. More statistics confirm this superiority in appendix.

### 5.4 Discrete benchmarks in Nevergrad

Our work essentially impacts the continuous optimization benchmarks. However, we present results associated to discrete benchmarks as depicted in Section 2.2. We observe good results for DiscreteLenglerOne-PlusOne, for example for the Bonnans benchmark (Bonnans et al., 2023) (Figure 2), which is completely different from the functions used for testing and designing DiscreteLenglerOnePlusOne that is mathematically derived on simpler functions. Results of DiscreteLenglerOnePlusOne are also good on InstrumDiscrete, SequentialInstrumDiscrete, and PBOReduced problems. In terms of ablation and robustness, we observe that most of the variants of DiscreteLenglerOnePlusOne with perturbed hyperparameters also perform well.

## 6 Discussions

In a reproducibility crisis, designing vast benchmarks that identify scaling issues is critical, and optimization wizards are a tool for aggregating the knowledge of the community in a single code.

**Multi-scale benchmarks.** We note that in both continuous and discrete benchmarks, the scale is important, even more specifically in the black-box case. For the continuous case, Figure 3 and Figure 6 and global statistics in Section 5.3 validate our approach (Cobyla as a warm-up step before other tools, or quasi-opposite sampling, or modern direct-search methods in the sequential case) for multi-scale benchmarks.

**In spite of (actually, even because of) the random shift method, many continuous benchmarks have roughly the same norm of the optimum for all instances leading to a poor evaluation of the "scaling" properties of the algorithms (in the sense: being robust to different scales of the distance to the optimum).** If we define the position of optima by e.g., a multivariate normal distribution with mean zero and identity covariance matrix (or more generally, independent coordinates with all roughly the same variance, so that variants of the central limit theorem can be applied), then in large dimension the optimum has, almost always, a norm scaling as $\sqrt{dimension}$ (see Section 4.2). This is not observed in real-world benchmarks, hence the great real-world performance of the methods above (quasi-opposite sampling) which tackle such issues. We advocate MS-BBOB or ZP-MS-BBOB for designing continuous artificial benchmarks close to scaling issues found in the real-world: their results are closer to real-world results (Section 5.2) than other artificial continuous benchmarks in the sense that quasi-opposite sampling

or a warm-up by Cobyla are helpful in both cases. More precisely, comparing Figure 5 (left, real-world) and Figure 3 (multi-scale): in both cases, the methods based on quasi-opposite sampling and warmup by Cobyla (such as Carola and NgIoh variants) perform well, though we also note independently (see Reality Gap below) good results for DE and PSO and their combination into NGOptRW in real-world settings.

**In the discrete case**, the best methods are frequently based on Lengler (Einarsson et al., 2019), which is based on a predefined schedule of mutation scales. This schedule differs from the classical $1/d$ mutation, in particular in early stages. We note that the mathematically derived Lengler method outperforms some empirically handcrafted methods in Nevergrad (with "BSO" in the name) based on the same principle of a decreasing rate, and many methods with adaptive mutation rates. It also frequently outperforms methods such as (Doerr et al., 2017b; Dang & Lehre, 2016), which use a fixed probability distribution of the mutation rates. We see the chaining of methods with different regimes in continuous domains as analogous to the predetermined schedule of (Einarsson et al., 2019) in the discrete case. In any case, both approaches perform well.

**Quasi-opposite sampling.** An unexpected result of our multi-scale work is the good performance of quasi-opposite sampling (Rahnamayan et al., 2007) (see QODE, QNDE, QOPSO, SQOPSO in Section 5.3). We adapted it from DE to PSO (as (Zhang et al., 2009)), with SQOPSO using, for each particle $p$ with speed $v$, another particle with position $-r \times p$ and speed $-r \times v$ (see Section 2.2). Equipped with quasi-opposite sampling, DE and PSO perform quite well in the real-world part of our benchmarking suite (Section 5.2 and Appendix G), with particularly good results of SQOPSO in the case of neuro-controllers for OpenAI Gym (confirmed in Figure 20). A posteriori, this is consistent with the importance of scale.

**Optimization wizards.** As in SAT competitions and as discussed in the Dagstuhl seminar (Hoos, 2023), we observe excellent results for wizards. All methods performing well on a wide range of benchmarks (without tuning for each benchmark separately) are wizards. NgIoh4 is based on NGOpt, a complex handcrafted wizard based on experimental data, and it adds insights related to multi-scaling from the present paper. It performs well on many benchmarks (Section 5.3). We see in the detailed logs that (similarly to NGOpt) it uses CMA, DE, PSO, Holland crossover, bandit methods for handling noise, discrete $(1+1)$ methods with mutation rates schedules, meta-models, Cobyla, multi-objective adaptations of DE, the simple $(1+1)$ evolution strategy with one-fifth rule (Rechenberg, 1973) in some high-dimensional contexts, bet-and-run, and others. NgIohTuned shows that NgIoh can still be improved by importing ideas from NGOptRW or quasi-opposite sampling, or by tuning its rules in favor of Carola2 or Carola3 in more general cases, and by incorporating DSproba instead of CMA in the sequential low-budget case. Appendix H.3 confirms that NgIoh4 and other wizards, besides outperforming NGOpt and non-wizard methods on many Nevergrad benchmarks, also outperform them on BBOB/COCO. NgIohTuned, using high-level information on the type of problem and the types of variables, outperforms all other wizards and aggregates in a single code all the conclusions in the present paper.

**Low-budget optimization (LBO), and first part of a chaining in continuous domains.** SMAC3 got better results than other Bayesian Optimization methods in many of our LBO experiments. Bayesian Optimization methods are often limited to low budget / low dimension contexts, and a strong competitor for LBO is Cobyla (Dufossé & Atamna, 2022; Raponi et al., 2023) which is computationally fast and frequently better even relatively to the number of iterations. Therefore, we propose to use Cobyla as a warm-up before other methods: it is good at understanding the global shape of a problem (Sections 3 and 5.1). Carola2 is a chaining of 3 stages: Cobyla for a fast first approximation, CMA with MetaModel for a robust optimization, and SQP for a final fast local search. Carola2 performs very well as a component of NgIoh4, and its counterpart Carola3 (compatible with parallel settings) performs well in many real-world benchmarks (Section 5.2). Chaining was already present in (Rapin & Teytaud, 2024), for both: (a) the classical fast local convergence at the end in many cases, and also (b) for noisy optimization, with a classical algorithm (not taking care of noise) as a first step before switching to a real noisy optimization method in the wizard of Meunier et al. (2022), but application of a mathematical programming algorithm as a warmup is rarer. Appendix H.3 shows that it is also valid for the BBOB/COCO benchmarks.

**Reality gap.** The gap between real world and artificial benchmarks is still large, as shown by the different best algorithms in real-world vs artificial contexts. In particular, in the continuous context, NGOpt/NgIoh4

dominate the artificial benchmarks whereas a bet-and-run (termed NGOptRW) of DE, PSO, and NGOpt is better in the real-world context. Also, quasi-opposite sampling appears to be great for the real-world context, more than for artificial benchmarks based on random shifts. Random shifts with all components of the shift being independent (or other methods than random shifts provided that many coordinates are independent and have roughly the same variance), lead to nearly the same norm of the optimum for all replicas. Our zero-penalized and multi-scale variants of BBO benchmarks, described in Section 5.1, do not suffer from the central limit theorem which makes the optimum to have always approximately the same norm, thanks to the random factor applied to all coordinates. They are therefore a step against the reality gap and we plan to add more of such benchmarks. Another important point in terms of bridging the reality gap is that including cases with budget far lower than the dimension is also essential (Ungredda et al., 2022).

An element related to the reality gap is that in most works on wizards, and in our proposed NgIoh4, the algorithm does not use information such that "this problem is real world" or "these variables are the weights of a neural net". Using such information is feasible thanks to the vast empirical results, and NgIohTuned leverages this information (Figure 5, right, and Appendix H). We note that in (AX-team, 2021; Awad et al., 2020), the best performing method was using the names of variables for choosing between the different options. Appendix H.2 presents results on real-world benchmarks with more details, confirming the gap between the global statistics (Section 5.3, dominated by NgIoh4) and the real world case (Section 5.2 and Appendix G, dominated by NgOptRW). The algorithm NgIohTuned (the only one among our methods which uses high level information about the problem provided by the user, such as "neuro-control" or "real-world"), modified for switching to NGOptRW in the real world case and to SQOPSO in the neuro-control case, performs best overall (Figure 5 (right) and Appendix H).

**Validating NgIohTuned.** We see in Figure 5 (right) that NgIohTuned (combining, by design, NgIoh4 in artificial benchmarks and NGOptRW in the real-world case and SQOPSO for neuro-control) outperforms NGOpt and all previous algorithms. Details: SODE, SQOPSO, QODE, QOPSO are defined in Section 2.2 and Appendix C. LargeCMA is CMA with greater initial variance. BAR is a bet-and-run of the $(1 + 1)$ evolution strategy and DiagonalCMA and OpoDE, where OpoDE runs the $(1 + 1)$ strategy with one-fifth rule during half the budget followed by differential evolution.

**Good benchmarks exist, with a lot of diversity (including real-world and artificial, budget $\gg$ dimension and budget $\ll$ dimension, with and without noise, with applications from completely different fields, see Table 1): they should be used, in particular in machine learning papers.** Reproducibility is a growing concern in machine learning (Kapoor & Narayanan, 2023), specifically in BBO (Markov, 2023): we confirm a necessary special emphasis on scaling discussed in (Meunier et al., 2022). In spite of efforts in the 2000s for creating better benchmarks, benchmarks with optimum at zero, or ad-hoc experiments with a heavily tuned method with parameters optimized *for each benchmark separately*, or with completely different initialization distributions for the baselines, are still published in many conferences.

## 7    Conclusions

We propose multi-scale continuous benchmarks, such as our MS-BBOB. They have the property that the distance between the center and the optimum significantly varies over the instances. We note the excellent performance of DE and PSO and bet-and-run on many real-world problems, with excellent results of quasi-opposite sampling on neuro-control problems. These two conclusions are used for deriving NgIohTuned, a black-box optimization wizard performing well on many benchmarks, in Nevergrad, in the wild, and in BBOB/COCO. All benchmarks are automatized and integrated into an open source benchmarking platform.

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

## A  Ablation regarding ZP and MS: the importance of scaling in continuous domains

We observe in Figure 3 good results for Carola2. MetaModel and several CMA variants are absent of the figure because we keep only the 25 best of the 57 tested methods: CMA, OldCMA (the previous version of CMA in Nevergrad before some tuning), LargeCMA (with larger initialization scale) and MetaModel (CMA plus a quadratic surrogate model, as used as a component of Carola2) are ranked 43, 29, 40 and 33 respectively (vs 3 for Carola2). We also tested several variants of Carola (Carola4+, available in the codebase and visible in some of our plots), without much difference.

Figure 6 presents more ablation results for selected algorithms.

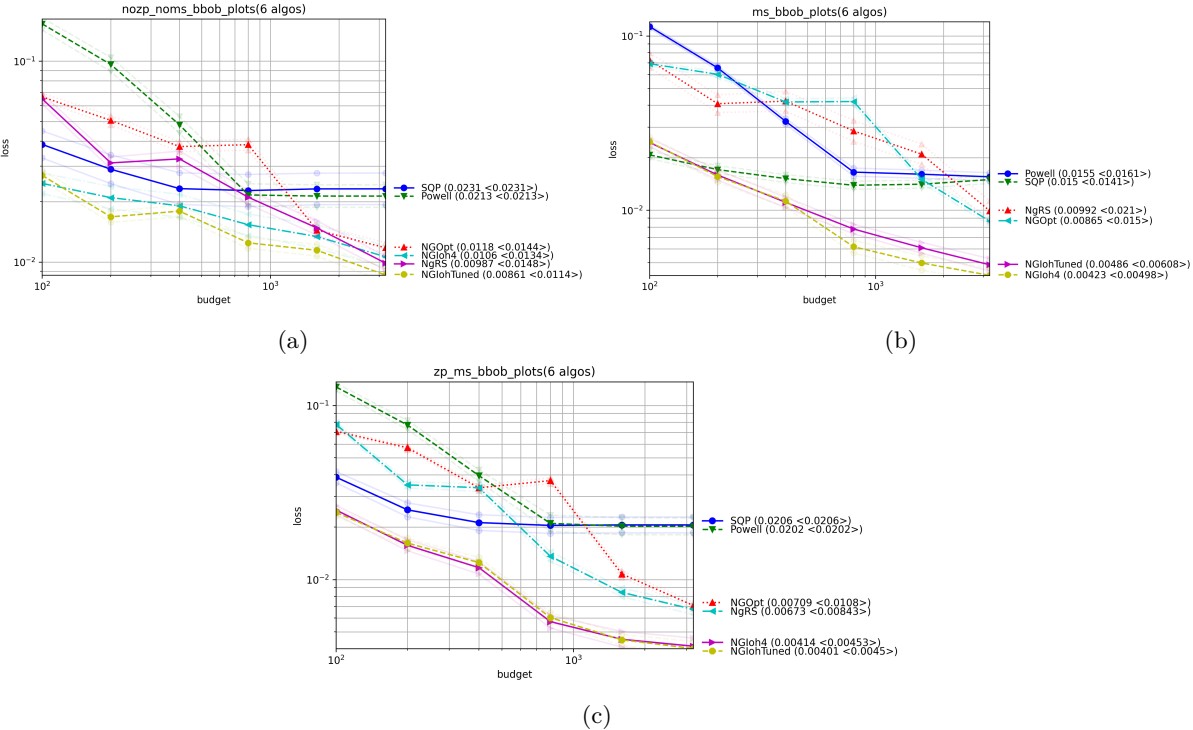

(a)                                                    (b)

(c)

Figure 6: Ablation for Figure 3 with the same functions and budgets in all three cases, and with a restricted set of algorithms (average normalized loss as in Section 4.3): (a) no ZP and no MS; second (b), we add MS; third (c), we add ZP. While ZP has less impact, we observe that switching from (a, vanilla) to MS (b, multi-scale) makes NgIohTuned or NgIoh4 vastly better than NGOpt, which is still the case after adding also ZP (c, MS, and ZP). We believe that such multi-scaling should be part of artificial benchmarks for bridging a part of the gap with real-world benchmarks. ZP has a smaller impact than MS.

## B  Reproducibility

How to reproduce the results in the present paper:

- Install Nevergrad by cloning the git repository (see details at (Rapin & Teytaud, 2024)).

- Running:
  - Without cluster: `python -m nevergrad.benchmark yabbob --num_workers=67` if you want to run YABBOB on 67 cores.
  - With cluster equipped with Slurm: Run "sbatch scripts/dagstuhloid.sh" script for launching experiments with Slurm. It is written assuming that Slurm is installed: it should be feasible

to adapt it to other job scheduling tools. Running this script several times will increase the number of replicas and increase precision.

- For plotting results, run "scripts/dagstuhloid_plot.sh". Some data might be missing if not enough runs are complete.

- To modify the parallelism, dimension, budget, list of tested algorithms, you might edit `nevergrad_repository/nevergrad/benchmark/experiments.py`.

The present paper in LaTeX is automatically generated by the commands above. Then, the authors have edited the created file for the text and rearranged sections, in particular, moving to the appendix or to an URL many of the individual results on specific benchmarks. An example of the huge original PDF file can be found at `tinyurl.com/dagstuhloid`. We emphasize that reproducibility is not limited to the possibility of reproducing the exact same numbers: results that can only be obtained for certain random seeds are less interesting. We therefore do not fix the seeds. NGOpt, and more generally Wizards in Nevergrad, contain many specific cases, e.g. for bounded problems or very low budget cases: we refer to (Rapin & Teytaud, 2024) for all details.

## C   Additional information on quasi-opposite algorithms

We use quasi-opposite DE, in several flavors:

---
**Algorithm 3** The QODE algorithm, with Curr-to-best, $F1 = F2 = .8$, $CR = .5$.

---
**Require:** Budget $b$, population size $p$ ($p = 30$ by default), dimension $d$, objective function $l$ to be minimized

    Randomly draw $p/2$ points (uniformly at random by default) $x_1, \ldots, x_{p/2}$.

    Define $x_{p/2+i} = -r_i x_i$, with $r_i$ randomly independently uniformly drawn in $[0, 1]$.

    Run differential evolution as usual:

    **while** Budget not elapsed **do**

        **for** each point $x$ in the population, if the budget $b$ is not elapsed **do**

            Randomly draw $a$ and $b$ in the population, different from $x$

            Let $t$ be the best point so far

            Define $x' = x + F1 * (b - a) + F2 * (t - x)$

            Define, for $1 \leq i \leq d$, $x''_i = x'_i$ with probability $CR$ and $x''_i = x_i$ otherwise.

            Enforce $x''_i = x'_i$ for some randomly drawn $i \in \{1, \ldots, d\}$

            If $l(x'') \leq l(x)$, then replace $x$ by $x''$.

        **end for**

    **end while**

---

- QODE, the classical quasi-opposite DE, presented in Algorithm 3.

- QNDE, which is QODE during half the budget and then BFGS with finite differences.

- SPQODE (SPecial QODE), which is QODE with population size $1 + \sqrt{\log(d + 3)}$ in dimension $d$.

- LQODE (Large QODE), which is QODE with initialization range multiplied by 10 (each individual is multiplied by 10).

- SODE (Special Opposite DE), in which $r$ is $\exp(-5 \times U(0, 1))$ instead of $U(0, 1)$ (with $U(0, 1)$ uniform in $[0, 1]$).

- QOTPDE combines TwoPointsDE (DE with Holland 2-points crossover) and QODE.

We also consider quasi-opposite sampling for PSO:

- Randomly draw half the population as usual.

---

**Algorithm 4** SQOPSO. By default, $p = 40$, $\omega = 0.5/\log(2)$, $\phi_p = 0.5 + \log(2)$, $\phi_g = 0.5 + \log(2)$.

---

**Require:** Budget $b$, population size $p$, dimension $d$, objective function $l$ to be minimized

Randomly draw $p/2$ points (uniformly at random by default) $x_1, \ldots, x_{p/2}$, and their speeds $v_1, \ldots, v_{p/2}$.

Define $x_{p/2+i} = -r_i x_i$ and $v_{p/2+i} = -r_i v_i$, with $r_i$ randomly independently uniformly drawn in $[0, 1]$.

Initialize $b_i = x_i$ for all $i$ ($b_i$ is the best past position of the $i^{th}$ particle and $g$ the best of the $b_i$

Run particle swarm optimization as usual:

**while** Budget not elapsed **do**

    **for** each point $x_i$ with speed $v_i$ in the population, if the budget $b$ is not elapsed **do**

        **for** each coordinate $1 \le j \le d$ **do**

            Randomly draw $r_p$ and $r_g$ in $[0, 1]$

            Update $(v_i)_j = p \times \omega (v_i)_j + \phi_p r_p (b_{ij} - x_{ij}) + \phi_g r_g (g_{ij} - x_{ij})$

        **end for**

        Update $x_i = x_i + v_i$

        If $l(x_i) < l(p_i)$, then update $p_i = x_i$

        If $l(x_i) < l(g)$, then update $g = x_i$

    **end for**

**end while**

---

- QOPSO (Quasi-Opposite PSO): for each point $p$ with velocity $v$ in this half population, also add $-r \times p$ with a randomly drawn velocity, with $r$ randomly drawn uniformly in $[0, 1]$.

- SQOPSO (Special Quasi-Opposite PSO, defined in Algorithm 4): for each point $p$ with velocity $v$ in this half population, also add $-r \times p$ with velocity $-r \times v$, with $r$ randomly drawn uniformly in $[0, 1]$.

## D Additional information on benchmarks

Table 1 discusses the diversity of the benchmarks in Nevergrad (left), and categorizes them (right).

Table 1: Diversity of our benchmarking platform in Nevergrad and of our automatic report.

(a)

| | Min | Max |
|---|---|---|
| Dimension | 1 | $20 \times 10^3$ |
| Budget | 10 | $3 \times 10^6$ |
| # objectives | 1 | 6 |
| Noise dissymetries | False | True |
| Noise | False | True$^\star$ |
| # blocks of variables$^\sharp$ | 1 | 16 |
| # of workers | 1 | 500 |

$^\star$many different levels of noise
$^\sharp$with independent rotations

(b)

| Category | Examples of benchmarks |
|---|---|
| Real-world, ML tuning | Keras, Scikit-learn (SVM, Decision Trees, Neural nets) |
| Real-world, not ML tuning | Gym, rockets, energy, fishing, photonics, games |
| Discrete | PBO, Bonnans, others (includes: unordered variables) |
| Continuous, artificial | LSGO, Deceptive, YABBOB |
| Multiobjective | Several problems with 2 to 7 objectives and dim from 2 to 200. |

## E Statistics over all benchmarks: full details

We point out that NGOpt and its variants are wizards (automatic algorithm selectors and combinators) created by the same authors as Nevergrad and their (good) results might therefore be biased: we recognize that common authorship for benchmarks and algorithms implies a bias, and, given that our tools are based on NGOpt and other tools in Nevergrad, this applies to us as well. Another issue is that statistics based on frequencies of performing in the top $k$ are a risky thing: when two codes are very close to each other, they are both penalized by each other: we must, therefore, be careful while interpreting the results. Nonetheless, we provide aggregated results for convenience (Figure 5, and details in the present section).

### E.1 NGOpt versus Base algorithms: validating wizards

Here base algorithms have no metamodel and no complex combinations: wizards are excluded, except NGOpt. NGOpt is the only sophisticated combination: this section is dedicated to validating NGOpt and validates that NGOpt performs better than the base algorithms it is built on. We consider statistics on the top $k$ methods, for $k = 1$, $k = 2$, $k = 3$: all results confirm the performance of NGOpt compared to base algorithms.

#### E.1.1 Number of times each algorithm was ranked first: NGOpt and base algorithms

- 29 NGOpt
- 8 HyperOpt
- 8 Cobyla
- 7 QODE
- 6 OnePlusOne
- 5 SODE
- 4 SPQODE
- 4 QOTPDE

#### E.1.2 Number of times each algorithm was ranked among the 2 first: NGOpt and base algorithms

- 45 NGOpt
- 16 QODE
- 16 OnePlusOne
- 16 Cobyla
- 12 HyperOpt
- 10 QORealSpacePSO
- 9 SQOPSO
- 8 QOPSO

#### E.1.3 Number of times each algorithm was ranked among the 3 first: NGOpt and base algorithms

- 51 NGOpt
- 23 Cobyla
- 20 QODE
- 20 OnePlusOne
- 19 SQOPSO
- 15 HyperOpt
- 14 QORealSpacePSO
- 12 SODE

### E.2 Comparing simple algorithms only: wizards, multi-levels, specific standard deviations, and combinations excluded

Simple algorithms might be less overfitted and more robust: we consider the same experiments as above but with only "simple" algorithms: no chaining, no metamodel, no tuned parameters, no bet-and-run, no wizard. The success (robustness) of quasi-opposite sampling (for PSO or DE) is visible. in the results below. We also note the excellent performance of Cobyla, thanks to great results for a moderate budget.

#### E.2.1 Number of times each algorithm was ranked first: no wizard, no combination

- 14 Cobyla
- 9 QODE
- 8 OnePlusOne
- 8 HyperOpt
- 6 QORealSpacePSO
- 5 SODE
- 5 QNDE
- 4 SPQODE

#### E.2.2 Number of times each algorithm was ranked among the 2 first: no wizard, no combination

- 17 QODE

- 17 OnePlusOne

- 17 Cobyla

- 13 QORealSpacePSO

- 12 SQOPSO

- 12 HyperOpt

- 10 GeneticDE

- 10 DiscreteLenglerOnePlusOneT

### E.2.3 Number of times each algorithm was ranked among the 3 first: no wizard, no combination

- 23 QODE

- 23 Cobyla

- 20 SQOPSO

- 20 OnePlusOne

- 16 QORealSpacePSO

- 15 HyperOpt

- 15 DiagonalCMA

- 13 OldCMA

### E.3 Everything included except NgIohTuned

For the results of this section, we include all codes, wizards as well as base algorithms. All strong methods are wizards, except tools based on quasi-opposite samplings. The only algorithms making it to the top are (i) wizards, (ii) bet and run/aggregations (such as SQPCMA), (iii) HyperOpt, (iv) quasi-opposite tools, and (v) Carola variants. We observe that the variants of NgIoh4 with different parameters (such as NgIoh2, NgIoh5, NgIoh6, or Wiz) do not bring improvements.

### E.3.1 Number of times each algorithm was ranked first: everything included

- 10 NgIoh4

- 9 SQPCMA

- 7 NgIoh6

- 6 NGOptRW

- 6 NGOpt

- 6 Carola3

- 5 Wiz

- 5 NgIoh5

### E.3.2 Number of times each algorithm was ranked among the two first: everything included

- 22 NgIoh4

- 14 NgIoh5

- 12 NgIoh6

- 11 SQPCMA

- 11 NGOpt

- 10 Shiwa (an old wizard, anterior to NGOpt, designed in Liu et al. (2020))

- 8 QODE

- 8 NgIoh2

### E.3.3 Number of times each algorithm was ranked among the three first: everything included

- 29 NgIoh4

- 27 NgIoh5

- 21 NgIoh6

- 16 Shiwa

- 14 NGOpt

- 12 HyperOpt

- 11 SQPCMA

- 11 QODE

## F   Additional experimental figures for artificial problems in Nevergrad

Figures 7 and 8 present results on variants of YABBOB with small ratio budget/dimension and LSGO. We observe a robust performance of NgIoh variants.

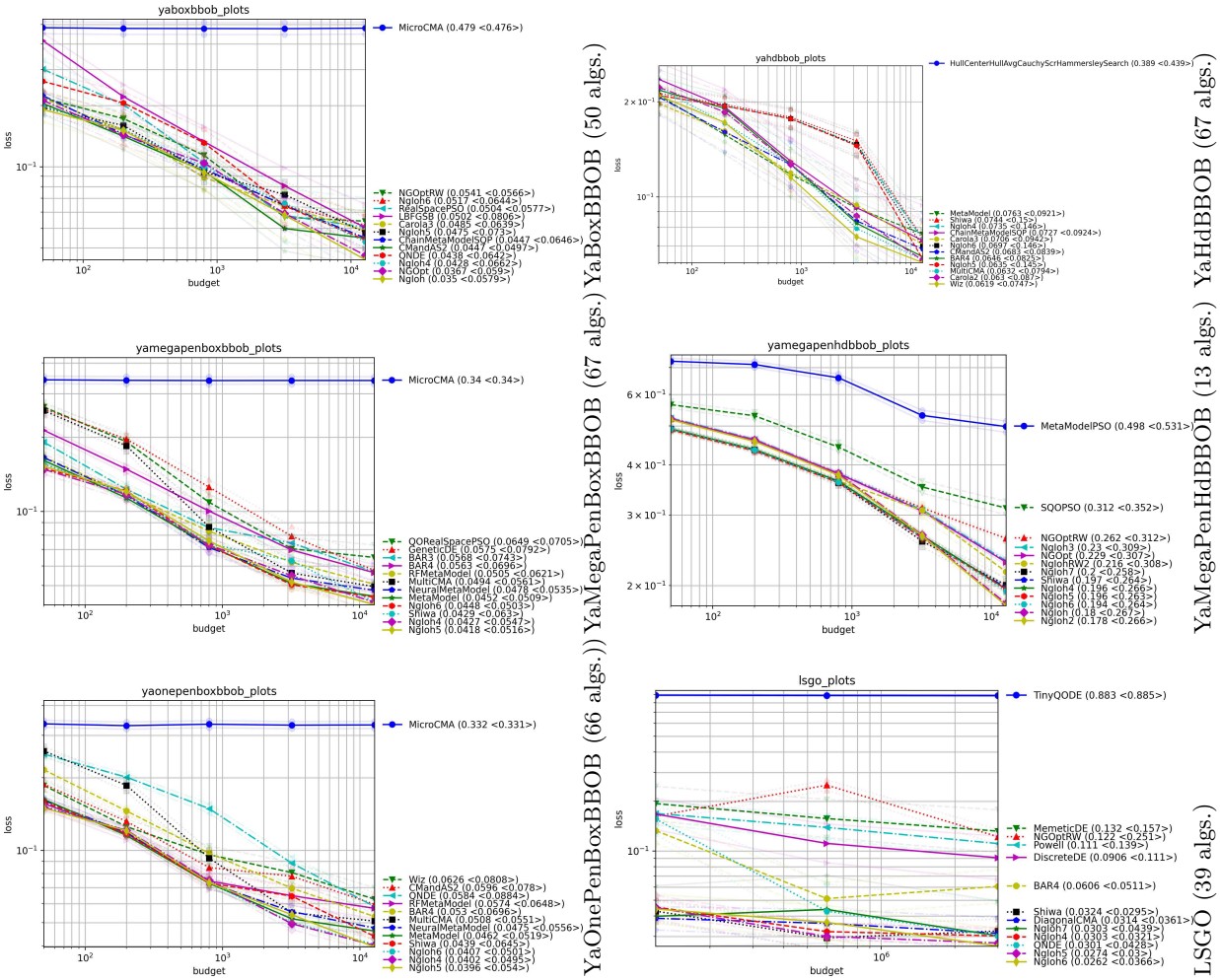

Figure 7: Validating NgIoh algorithms on variants of YABBOB with small ratio budget/dimension and LSGO. Other variants of BBOB are presented in Figure 8. This is the average normalized loss (see details in Section 4.3), with only the best methods (NgIoh4 is always there) and the single worst; see Figures 9 to 11 for more methods in the frequency of winning figures. On the right hand side: name of benchmark and number of algorithms run.

Figures 9 to 11 present results on variants of YABBOB in a format presenting more (though not all) algorithms. NgIoh variants (and to a lower extend wizards including Wiz and Shiwa and CMandAS2) perform well.

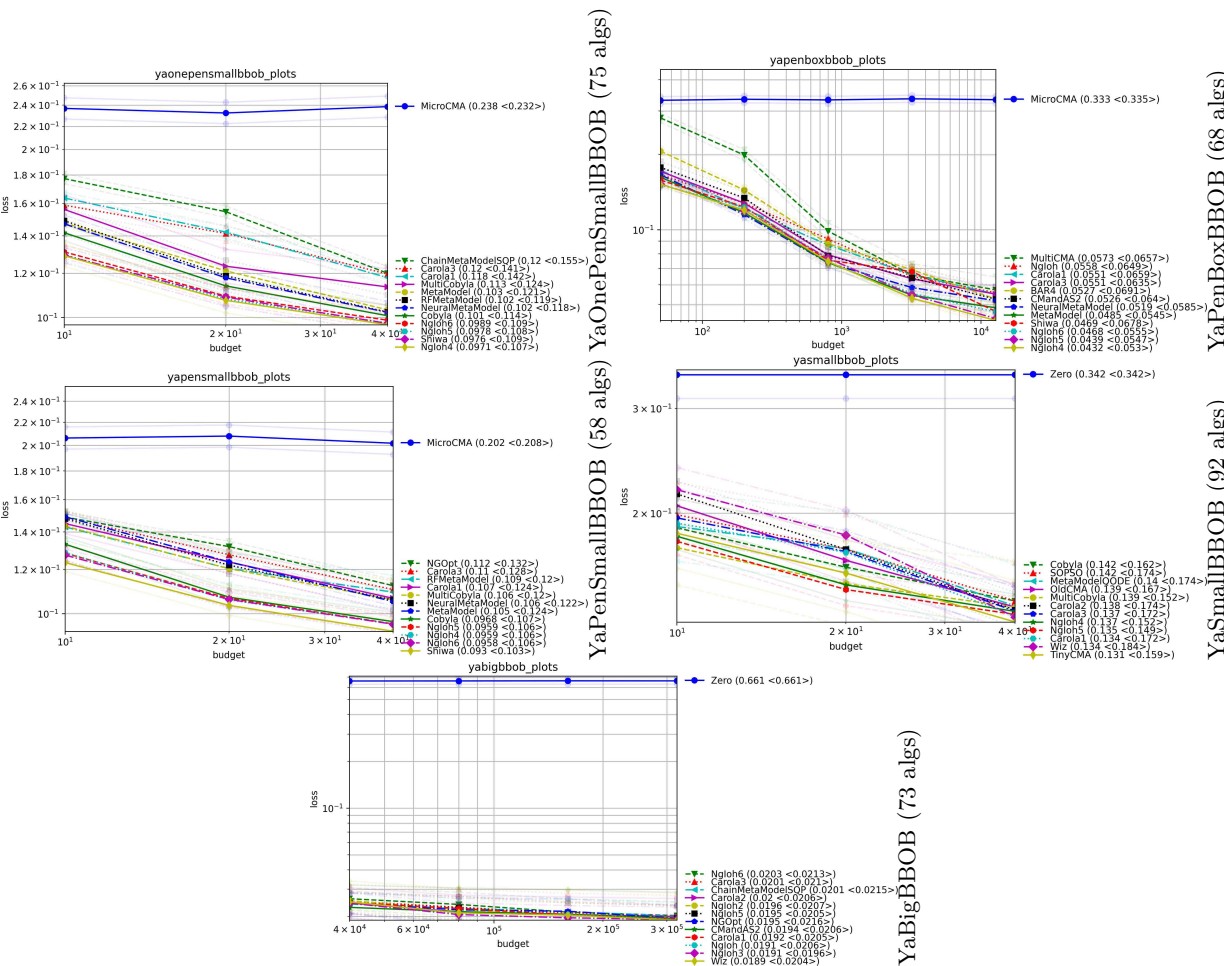

Figure 8: Validating NgIoh algorithms on other variants of YABBOB with small ratio budget/dimension. The last one, YaBigBBOB, is the opposite, with a large ratio budget/dimension. Only the 12 best methods and the worst are presented, all benchmarks include several variants of CMA, DE, PSO and others (see referenced URLs or Figure 11 for all details and more algorithms). Overall, NgIoh variants are excellent.

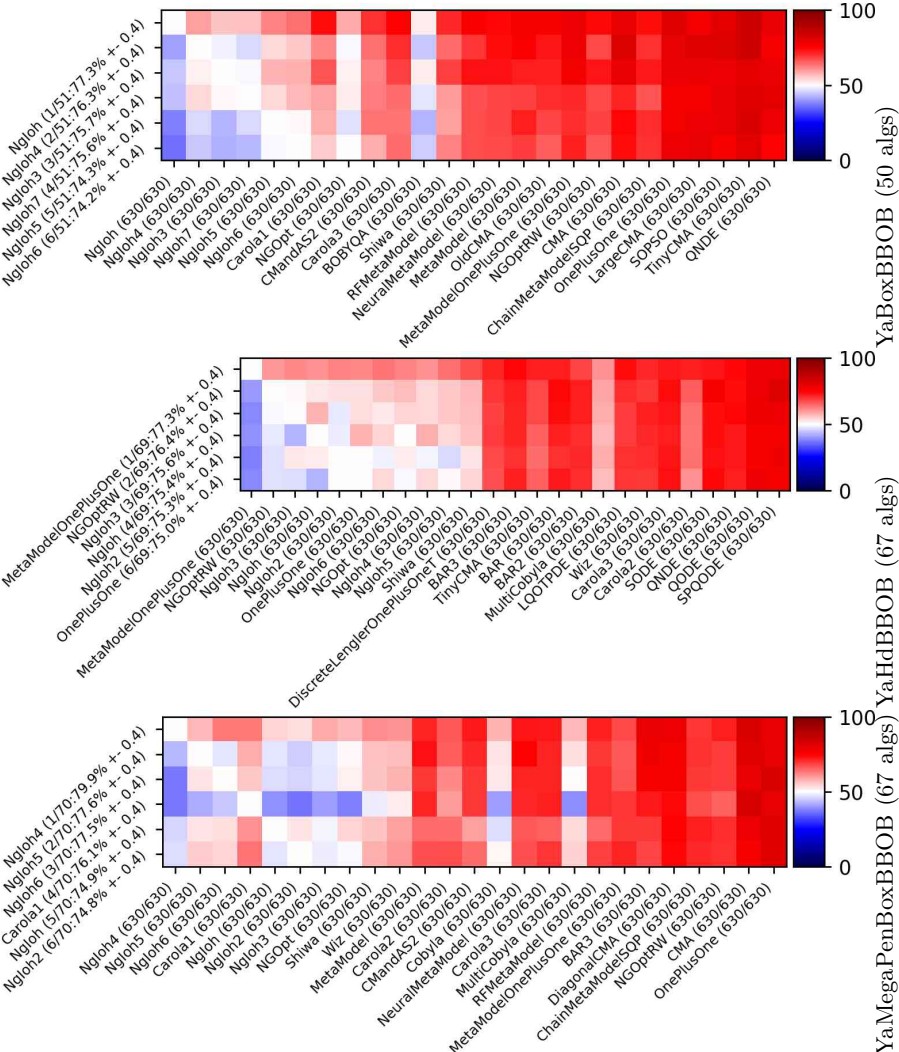

Figure 9: Validating NgIoh algorithms on variants of YABBOB with small ratio budget/dimension (different presentation than Figure 7). Other variants of YABBOB in Figures 10 and 11. This is the frequency of winning figure (see the detailed setup in Section 4.3), with the best methods on the left. NgIoh variants dominate.

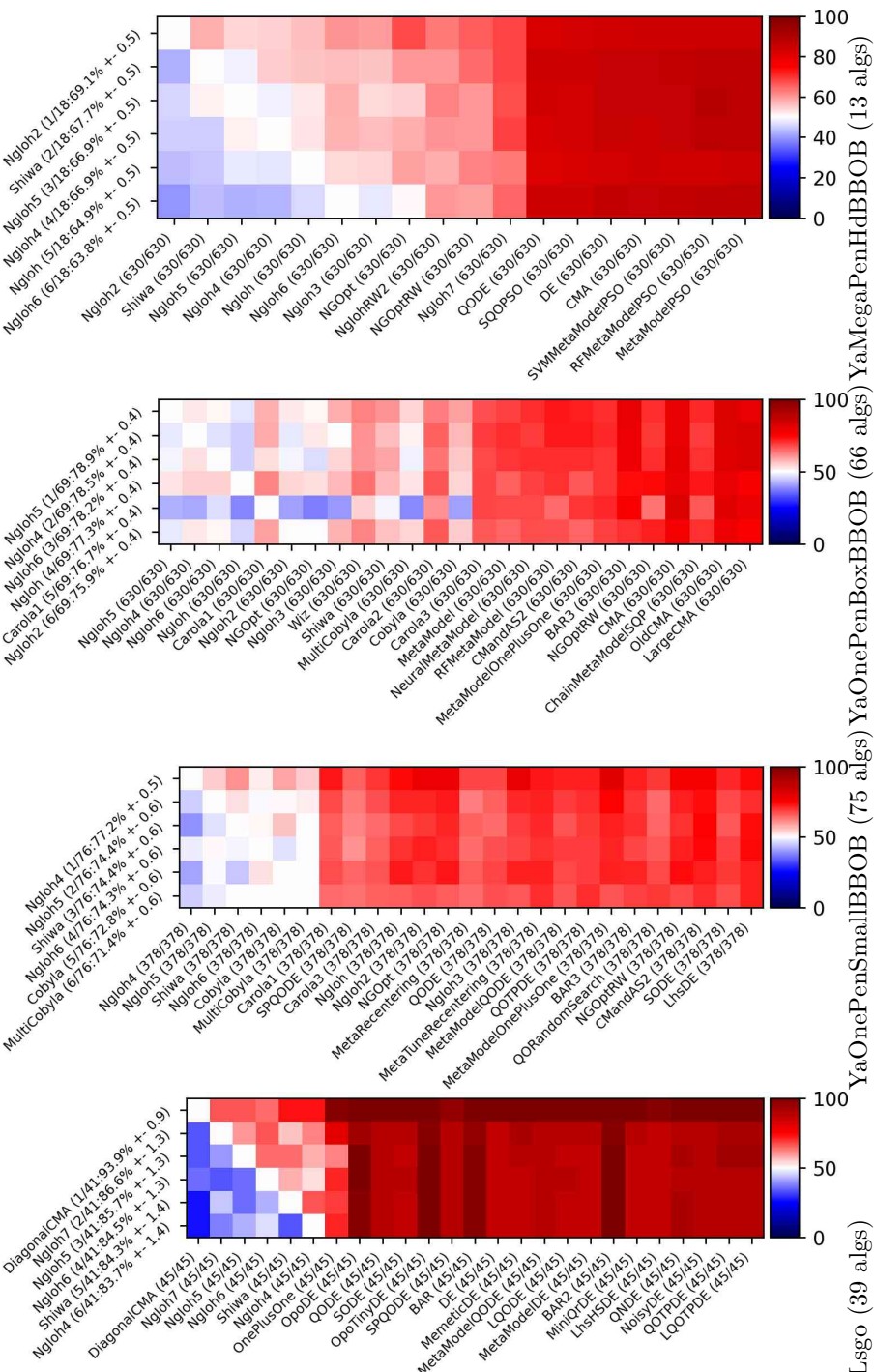

Figure 10: Validating NgIoh variants on other variants of YABBOB with small ratio budget/dimension and LSGO (Li et al., 2013) (different presentation than Figure 7). Other variants of YABBOB in Figure 9 and Figure 11. This is the frequency of winning figure (see details in Section 4.3), with the best methods on the left. Total number of methods run on the right.

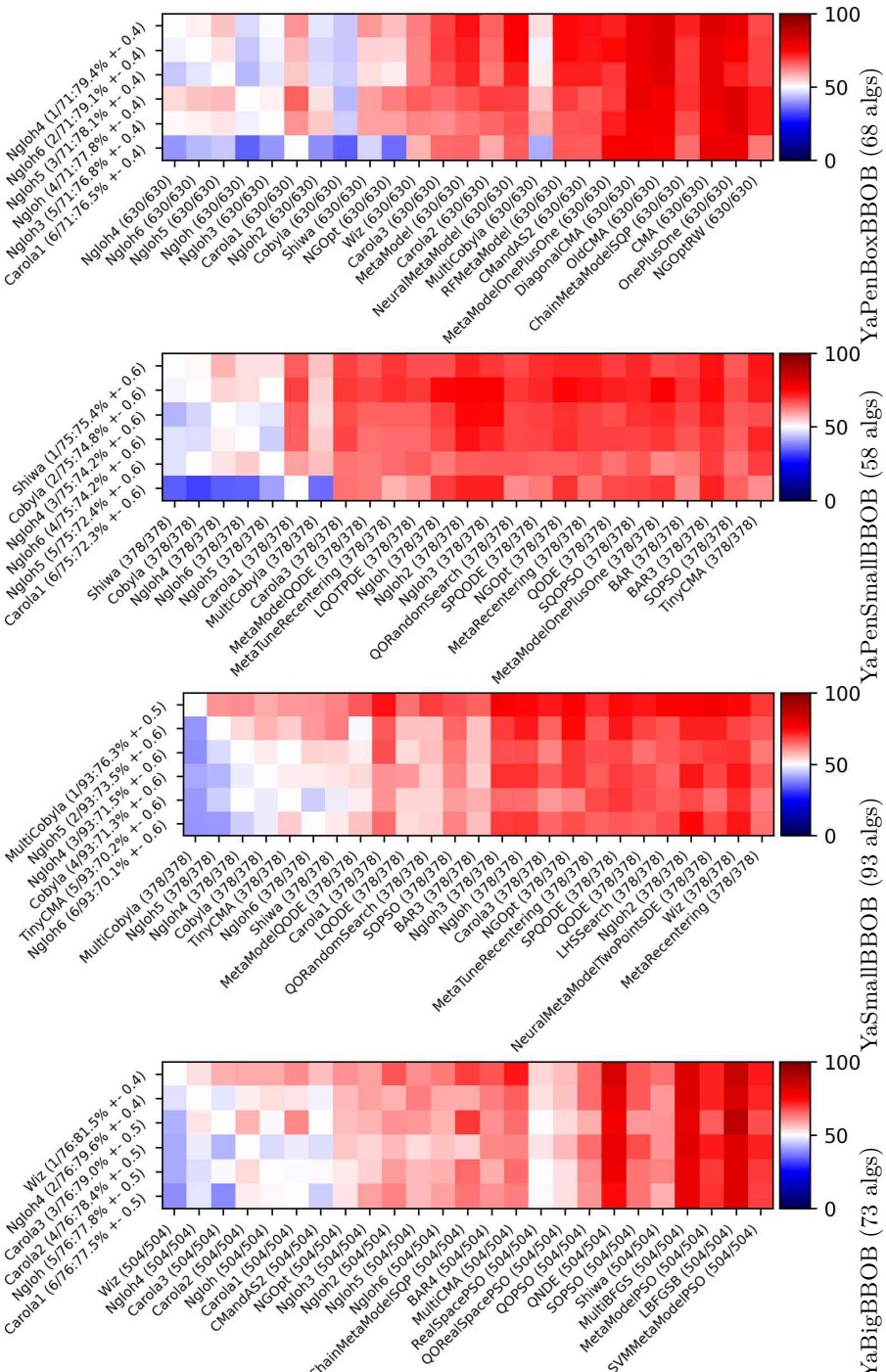

Figure 11: Top: 3 variants of YABBOB with small ratio budget/dimension (different presentation than Figure 8). Bottom: YaBigBBOB, is the opposite, with a large ratio budget/dimension. Frequency of winning figure as detailed in Section 4.3, with the best methods on the left. NgIoh variants and (except for the last) Cobyla dominate.

# G  Outside Nevergrad: application to external real-world problems

Finally, we include a few use cases by Nevergrad users. The benchmarks and setups have been developed independently of the benchmarking platform included in Nevergrad. The plotting tools, functions, and criteria, are frequently different from the rest of the paper. They, on purpose, quantify the robustness of the conclusions drawn on our update of the Nevergrad benchmark, specifically for the real-world cases. Overall, results in Appendices G.1, G.2, G.4 and G.6 confirm the conclusion, in Nevergrad benchmarks, that DE performs well on many real-world problems; the discrete problem in Appendix G.3 confirms the good performance of Lengler though FastGA (Doerr et al., 2017b) is also good; Appendix G.5 confirms the performance of SQOPSO when the scale of the optimum is unknown, in particular in the neuro-control case.

## G.1  Infrastructure: optimizing a caching policy

In this application, Nevergrad is used to optimize a caching strategy. The problem comprises 84 decision variables for the optimization. These variables encode the cache strategy. We run each method in several variants, with random parameters $a$, $b$, and $c$ so that constraints are penalized by $a \times constraintViolation^b \times i^c$, with $i$ being the iteration index. With this dynamical constraint penalization scheme, constraints violations are increasingly penalized so that eventually solutions without any violations are found. Compared to artificial benchmarks above, the setting has been influenced by the computational cost. All methods including GeneticDE, PSO, DE, TwoPointsDE, DiagonalCMA were run the same number of times, and the 11 best results are presented. We observe (Figure 15, left) that GeneticDE performs best and in general, DE variants perform well. One of the conclusions from this experiment is how much most Bayesian methods cannot be used for large budgets and dimension spaces larger than 84 (none of the methods available in Nevergrad was usable here), and computing gradients by finite differences (introducing a factor 85 in the computational cost) is also unfeasible. The results are consistent with the effectiveness, in our benchmarking suite, of DE variants for real-world problems with similar size/budget (Section 5.2). However, we would not have guessed the good performance of GeneticDE for this specific problem. Another observation is that we get a strong improvement compared to the handcrafted heuristic implemented before using the standard algorithm ($+70\%$) and also better than the manually designed solution (initial point). The problem is repeated: there are frequently new versions to be solved, so that doing this experiment is useful for doing a choice of algorithm for the future. We note (unpresented experiments) that Inoculation (Inoculation, here, consists in adding in the population 8 points obtained in previous optimization runs) roughly reduces the computational cost by a factor of five. We get roughly the same performance with 20% of the budget.

## G.2  Crop optimization

This application combines Nevergrad, PCSE (de Wit, 2021), and NASA data (Sparks, 2018) for optimizing the choice of crops in many countries. Figure 12 presents a specialization of the code to Kenya, including choosing crops and their varieties, depending on climate. Compared to the original code in Nevergrad, there are additional variables, for choosing the crop and the variety. After the present performance check (confirming the good behaviour of NGOptRW), a forthcoming publication is under work for various crops and continents.

## G.3  Mobile Network Base Station Placement Optimization

Figure 13 presents the experimental results regarding the optimization of the placement of base stations for a mobile network. An original ad-hoc implementation already existed before testing Nevergrad on this problem. The method which typically performs best in our discrete benchmarks, namely Lengler (Doerr et al., 2019; Einarsson et al., 2019), which uses a fixed, predefined mutation schedule and FastGA (Doerr et al., 2017b), which is also a method with a fixed mutation schedule, but here the schedule is a stationary stochastic random variable. We observe that while methods in Nevergrad perform well for low budget and outperform the original method by far, the original method performs best for greater budgets. Seemingly, the key point is that it uses specific mutation operators, whereas Nevergrad focuses on lists of variables with

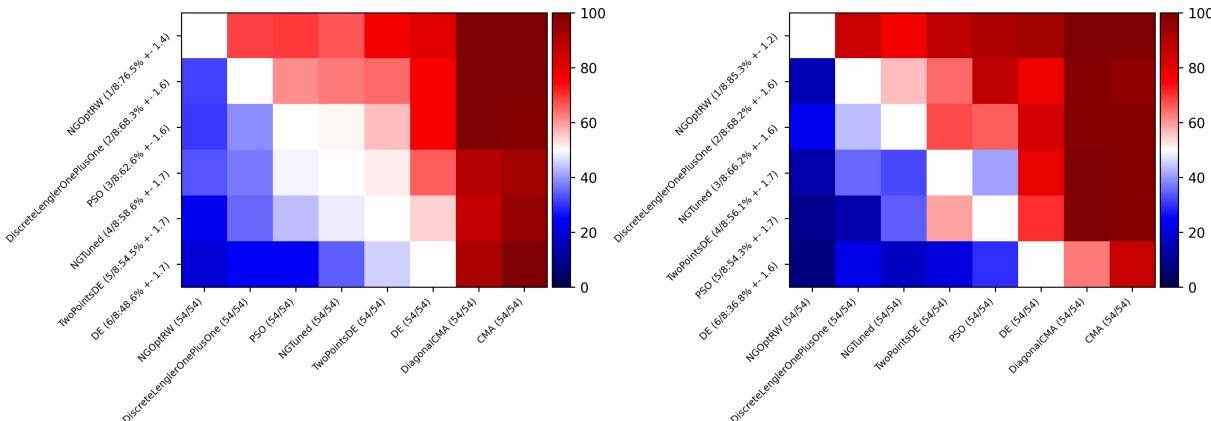

Figure 12: Comparison between optimization methods for crop optimization in the case of Kenya (left: 2011, corresponding to a particularly dry year; right: 2006). Setup as in Section 4.3: the heatmap shows the frequency at which method X (row) outperforms method Y (col). Rows and cols are ranked by average frequency against all other methods: top/left is best. As in many real-world cases, NGOptRW is excellent.

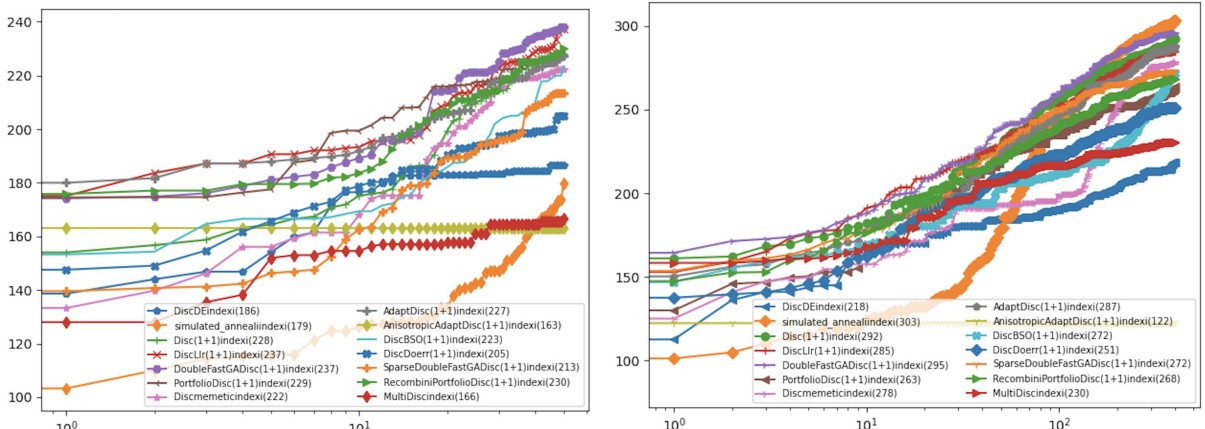

Figure 13: Placement of base stations of a mobile network: optimization with budget 50 (left) and 400 (right): the greater, the better, average best score between parentheses. We observe that Nevergrad methods performed quite well for the low budget case, but the specific method (Simulated Annealing with ad hoc mutation operator, in orange) developed for the problem at hand is the best for budget 400. Between parenthesis, the best obtained value. Llr is short for Lengler (Doerr et al., 2019; Einarsson et al., 2019).

generic operators. Nevergrad improves results on this 200-dimensional problem when the budget is 50, but not with budget 400.

## G.4 Robust topology optimization

Figure 14 presents the results for the optimization of mirrors smaller than a micron aimed at reflecting light at wavelength between 400nm and 650nm using only two materials.

Only the 7 best performing methods are presented, but actually 30 methods are run: There are 5 algorithms: DE, BFGS, Chain (a chaining of DE during half budget, followed by BFGS) from Nevergrad and the DE and Chain from Pymoosh (Langevin et al., 2023). For differentiating methods from Nevergrad, we add a prefix Ng for those methods. Each of them is run with a sampling parameter in $\{-100, -60, -20, 20, 60, 100\}$, hence 30 methods. This parameter specifies how robustness to wavelength is taken into account and has little impact here. The detailed description is beyond the scope of this paper. Another sampling parameter is

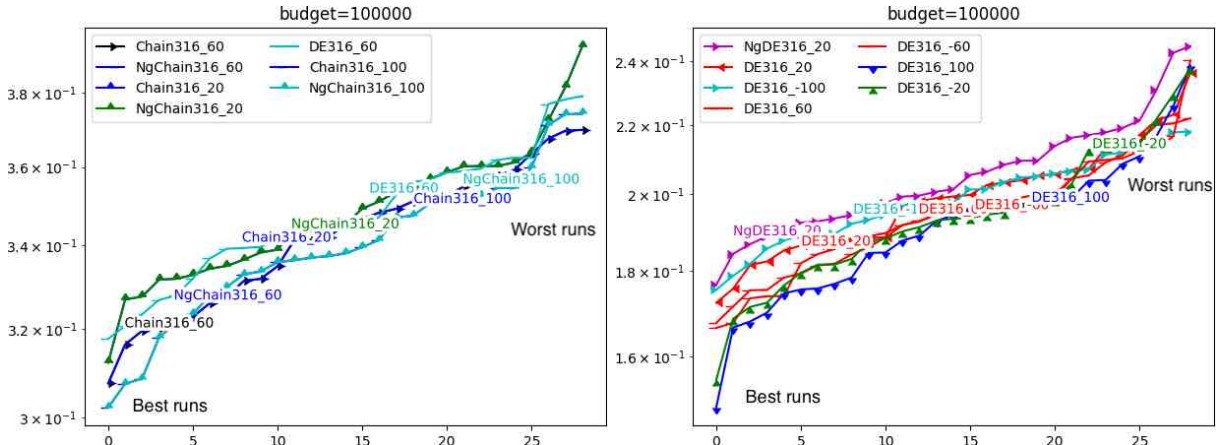

Figure 14: We perform photonics optimization (mirrors for various wavelengths) for 40, 50, 60, 70, 80 layers respectively: we keep only 40 and 80 for short and refer to the appendix for more. For each algorithm, we plot the results of 30 runs (best on the left, worst on the right): a horizontal curve means constant results, whereas a sharp increase means variable results. Methods are ranked by median value. Only the few best are presented for readability (legend: best at the end, i.e. bottom of the right column): extended version with more algorithms in the ranking in appendix, Figure 16. For moderate numbers of layers, the ranking is unclear (with Chaining of DE and BFGS frequently good), whereas for large-scale versions, DE dominates (for 80 layers, the 6 codes based on DE corresponding to the 6 values of the sampling parameter dominate all 24 other codes). The impact of the sampling parameter (suffix of the algorithm name) is unclear.

fixed at 316 (the square root of the budget) after preliminary experiments: it is actually the most important choice in the optimization design, other values are removed from plots as this is not the point in the present paper.

We observe that all strong methods, in the highest dimensional cases, are DE (either the one from Nevergrad, which is quite standard, or the one in PyMoosh which has been optimized for the problem at hand). This limited comparison validates the choice of DE in PyMoosh, though testing more algorithms could be possible. In lower dimension, we observe that the chaining of DE and BFGS frequently performs better than DE or BFGS alone.

### G.5 Gym

Nevergrad contains OpenAI Gym problems, which were deprecated after the issues of Gym v0.24.0, so that Gym was not included in recent exports of the Nevergrad benchmarks. We update the code importing Gym and rerun the experiments. Our code is merged in the codebase. Results are presented in Figure 15 (right): SQOPSO performs well. We observe an excellent performance of GeneticDE here, though this requires further investigation: results are excellent for some benchmarks and not for others, as opposed to quasi-opposite sampling which performs very well on most real-world problems, or as opposed to Carola2 and its integration in the NgIoh4 wizard defined in Section 3, which performs excellently on many benchmarks as discussed later.

### G.6 Photonics

Figure 16 presents results on photonics optimization, validating the performance of DE (part of the real-world bet-and-run used in NGOptRW) for this real-world problem, in particular in the higher dimensional cases.

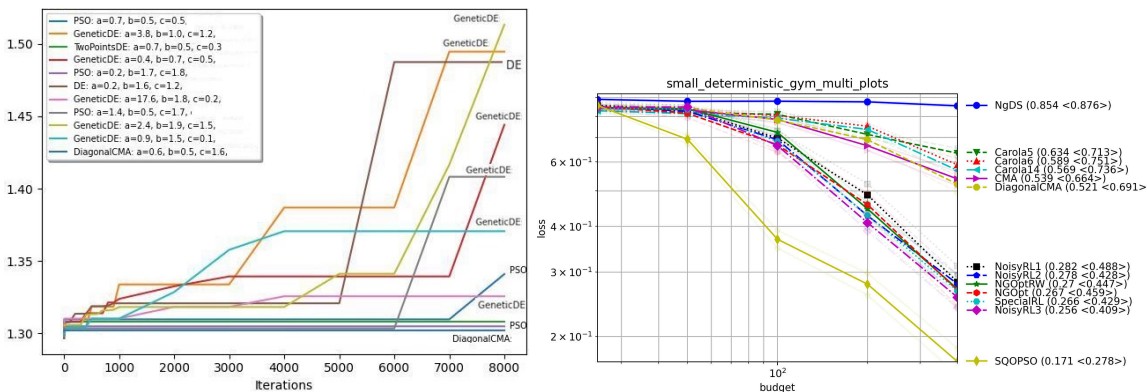

Figure 15: Left: Comparison of various methods on the infrastructure problem. The upper the better, only the 11 best results are presented: though on many benchmarks it was not so good, we note that runs based on GeneticDE are frequently among the 11 best, whereas all methods were run the same number of times. Right: Experiments on Gym (more algorithms and more benchmarks in Figure 20), confirming the good performance of SQOPSO (and existing wizards dedicated to reinforcement learning, with RL in the name (Rapin & Teytaud, 2024)) for neuro-control.

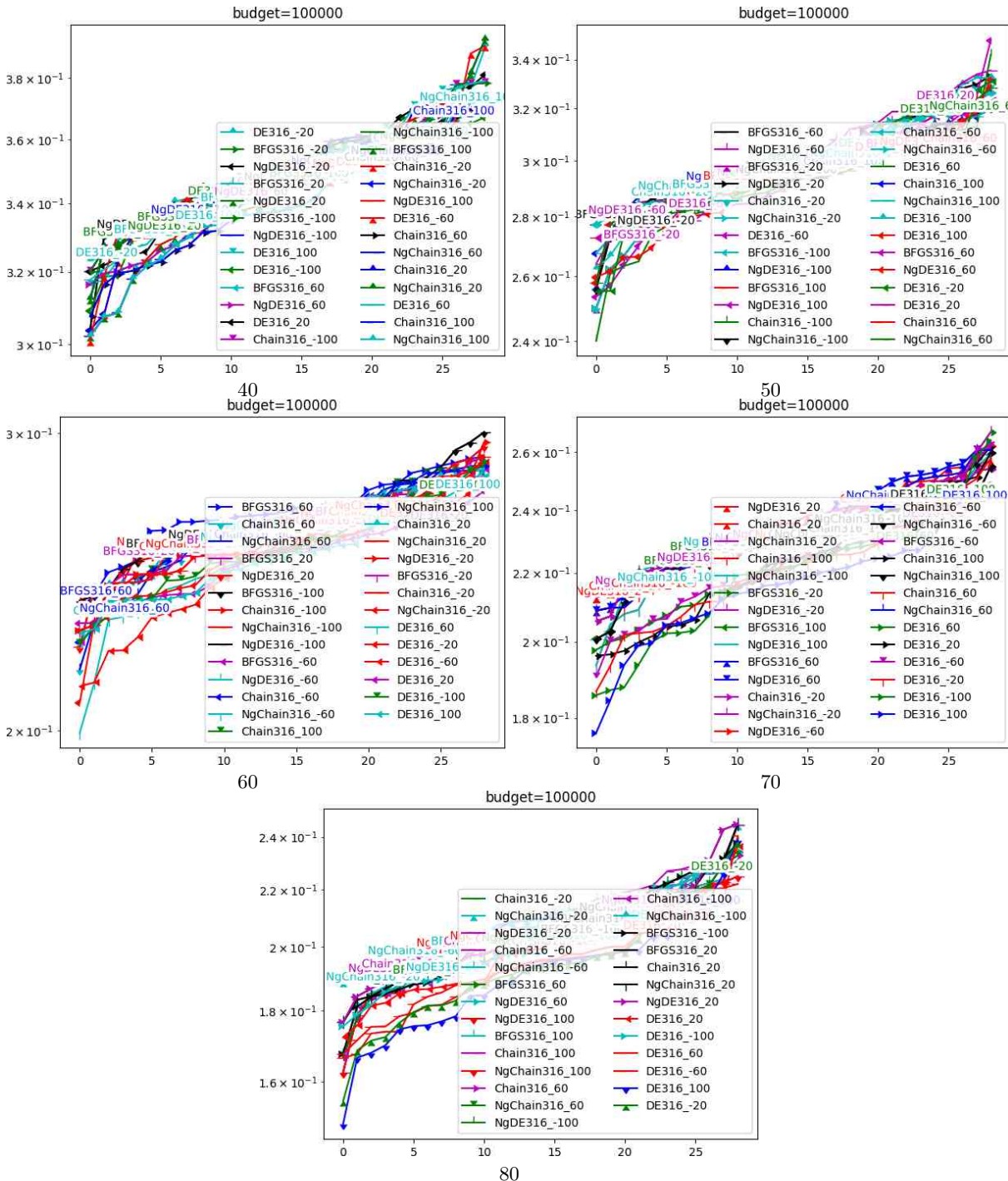

Figure 16: Photonics optimization (mirrors for various wavelengths) for 40, 50, 60, 70, 80 layers respectively: sorted result of the 30 runs of each method (best run on the left and worst run on the right). The 27 best (for the median) are presented (best at the end, right column, bottom), this extends Figure 14. For moderate numbers of layers, comparisons are unclear, whereas for large-scale versions DE dominates.

# H  Additional results with more variants

In the present section, we compare several new variants of NGOpt, including

- variants of NgOpt based on new direct-search methods from (Roberts & Royer, 2023) (see Section 2.2), other than NgIohTuned, with NgDS in the name. These variants derive from our NgIoh but use the direct-search method DSProba from Roberts & Royer (2023) instead of CMA in sequential cases.

- variants of NgOpt based on LogNormal mutations, such as NgLn, which are based on NgIoh but with Cobyla replaced by LogNormal mutations (Kruisselbrink et al., 2011b) as a warmup step.

The present appendix also compares SQOPSO and NgIoh4 and two more algorithms based on them:

- NgIoh21 (name: NgIoh "to 1", bringing all ideas into one single NgIoh) is NgIoh4 with constants within Carola2 modified (10% for Cobyla, 80% for the CMA with MetaModel, 10% for the final convergence with SQP), and replaces CMA by DSproba in the sequential case with budget lower than the dimension. The difference with NgIohTuned is that it does not leverage high level information such as "this is a real-world problem" or "this is a neuro-control problem".

- NgIohTuned (available in Rapin & Teytaud (2024)) is similar to NgIoh21 but it uses (i) NgDS instead of NGOpt in sequential cases (ii) VLPCMA (i.e. CMA with population size multiplied by 100) instead of CMA when the budget is greater than 2000 times the dimension (iii) most importantly, additional information provided by the user (if any) for switching to different algorithms: it switches to SQOPSO for neural control and to NGOptRW for other real-world problems.

- SQOPSODCMA, a chaining of SQOPSO (half budget) and Diagonal CMA (second half of the budget).

Basically we confirm with results below that NGOptRW (using a lot of DE and PSO) is preferable to NGOpt in many real-world contexts, that SQOPSO is better in the neuro-control case, and NGOpt variants using a first exploration step by Cobyla (as our method NgIoh4 does; LogNormal in lieu of Cobyla can also lead to interesting results) perform better than NGOpt in particular for benchmarks such as MS-BBOB and ZP-MS-BBOB which carefully control for the norm of the optimum.

## H.1  Additional results: YABBOB, ZP-MS-BBOB, MS-BBOB

We observe in Figure 4 and Figure 17 that results on YABBOB are mixed (NgIoh variants sometimes better than NGOpt and sometimes worse), whereas for MS-BBOB and ZP-MSBBOB all strong methods use either Cobyla or quasi-opposite sampling as a first step, validating our contributions:

- MS-BBOB: NgIoh4, SQOPSODCMA, NgIohTuned, NgIoh21, SQOPSO, all outperform all other methods, including NGOpt.

- ZP-MS-BBOB: NGIoh4, SQOPSODCMA, NgIoh21, NgIohTuned, all outperform all other methods, including NGOpt.

We note that the succesful codes (outperforming NGOpt on multi-scale benchmarks) are exactly

- the codes using quasi-opposite sampling as a first stage (as SQOPSODCMA, in both benchmarks) before using a local convergence method,v and

- the codes which use our chaining initiated by Cobyla (NgIoh variants),

except, in one of the two benchmarks only, SQOPSO, which still has quasi-opposite sampling but no second stage. By contrast, all other codes, except SQOPSO for one of the two benchmarks, perform worse than NGOpt. These results confirm the relevance of quasi-opposite sampling or Cobyla as a first stage for problems equipped with multiple-scale such as MS-BBOB or ZP-MS-BBOB.

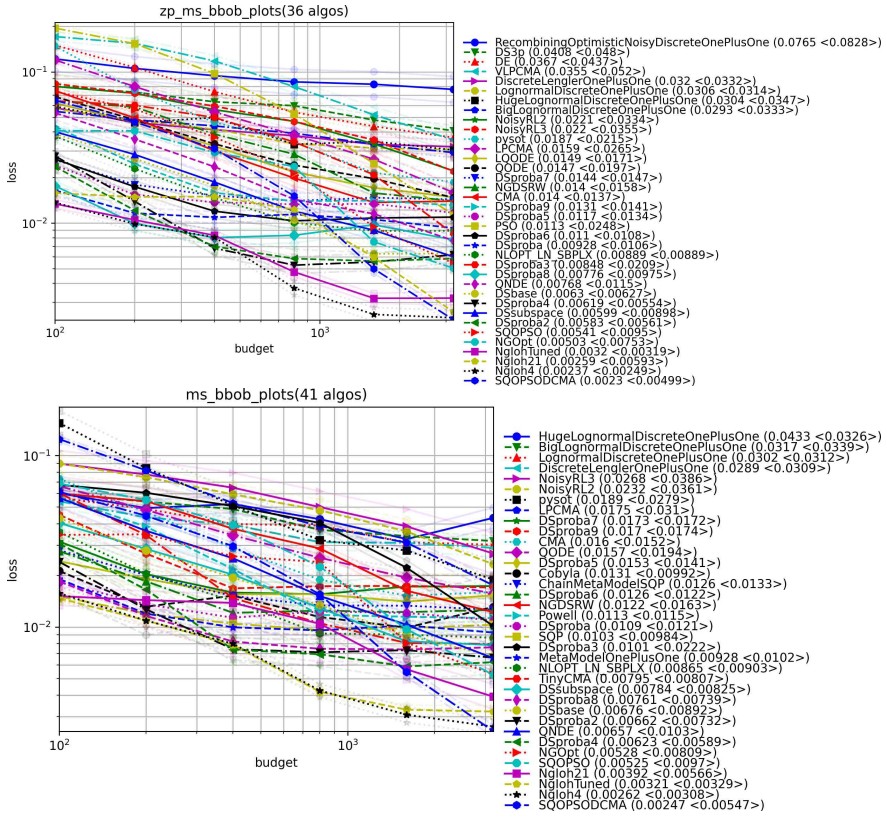

Figure 17: Additional results on ZP-MS-BBOB, MS-BBOB with more algorithms (see Figure 4 for the case of YABBOB, i.e. without the multi-scale approach of ZP-MS-BBOB and MS-BBOB). NGOpt is competitive on YABBOB (and our methods are not always at the top, though they are good), but fail (and many methods as well) compared to our methods in the cases of MS-BBOB and ZP-MS-BBOB. This shows how much results are different when we consider multi-scale benchmarks. Note the good performance of NgIoh4 and NgIoh21 and NgIohTuned on ZP-MS-BBOB and MS-BBOB for most budgets.

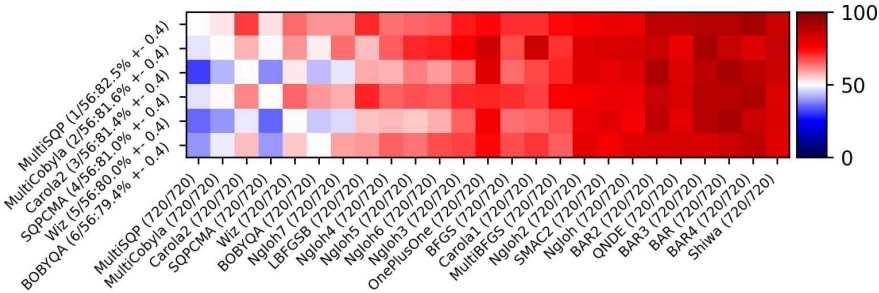

Figure 18: Counterpart of Figure 3 for ZP-MS-BBOB with the frequency of winning (26 best, out of 58 methods) instead of the normalized average loss. This takes into account all budgets as detailed in Section 4.3, hence the ranking is not the same as in Figure 3 which ranks only based on the results for the maximum budget, We note that many mathematical programming methods (using SQP, Broyden–Fletcher–Goldfarb–Shanno (BFGS) with finite differences, BOBYQA or Cobyla) are excellent for low budget, which corroborates the idea of using Cobyla as a first step in Carola2. 58 algorithms are run (the 26 best are presented), and the previous wizard, NGOpt, is ranked 22 and CMA is ranked 38 (this differs from statistics on normalized average loss as in Figure 3 but we still observe a superiority of NgIoh methods compared to CMA or NGOpt, a conclusion which is not so clear on non multi-scale benchmarks such as e.g. YABBOB).

## H.2 Additional results: real-world problems from Nevergrad

Figure 19 presents results of many algorithms on Aquacrop. We observe that on this real-world problem, all successful algorithms use DE or PSO, confirming the excellent behaviour of these algorithms in such settings. Also, the top algorithms frequently use either quasi-opposite sampling or GeneticDE (as in NoisyRL3, a wizard specifically designed for reinforcement learning), both aimed at taking care of the scale (NoisyRL3 and NGDSRW use GeneticDE, LQODE is a DE with quasi-opposite sampling, SQOPSODCMA uses quasi-opposite samping).

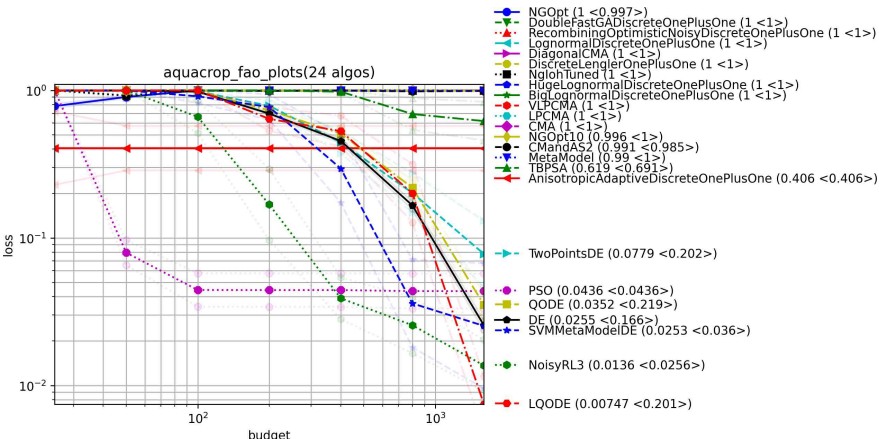

Figure 19: NGOpt is outperformed on the Aquacrop problems by all algorithms based on DE or PSO, even more when these algorithms use quasi-opposite sampling.

Figure 20 confirms the good performance of SQOPSO for neural control in deterministic contexts, with the application to Gym.

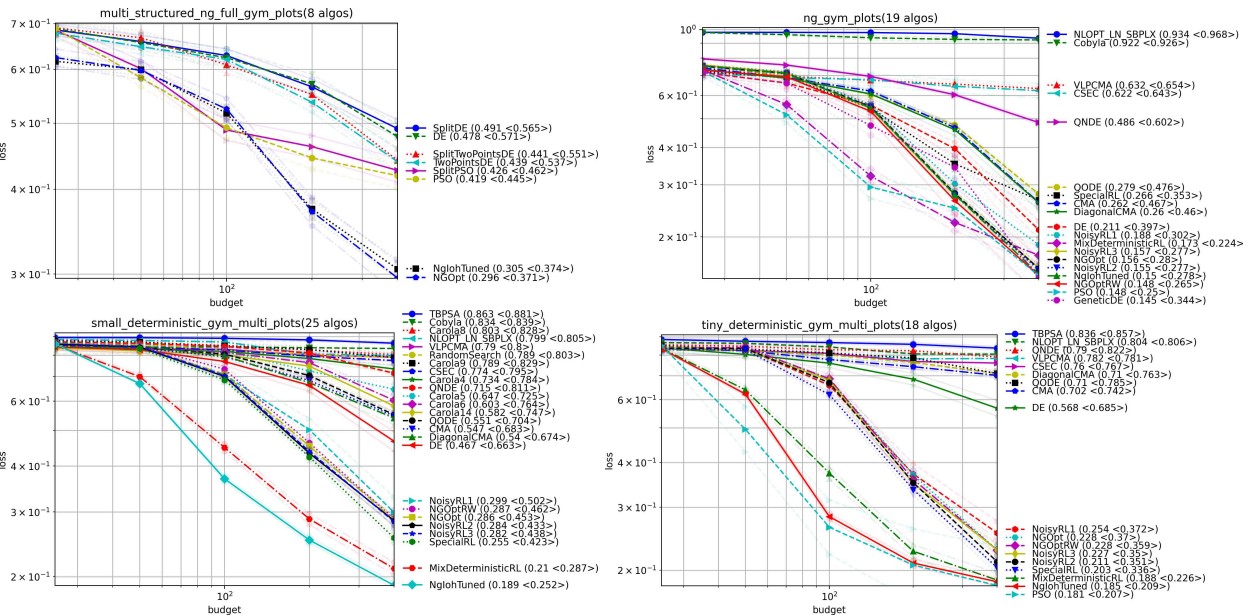

Figure 20: In the case of neural control, NgIohTuned (which uses the knowledge that this is a neural problem) switches to SQOPSO, hence its excellent performance. This figure confirms the excellent performance of SQOPSO for neural control.

### H.3 Additional results comparing NgIoh to NGOpt on the old BBOB/COCO

We compare

- NgIoh4,

- NgIoh21 which is a more recent version of NgIoh4 with a tuning of constants of the chaining (10%, 80% and 10% instead of three thirds in Carola2) and increasing the population of CMA by a factor 100 (compared to the default) for budget greater than 2000 times the dimension,

- NgIohTuned, a more sophisticated improvement based on NgIoh21, but also high level information such as "is a real-world problem" or "is a neuro-control problem" for selecting NGOptRW and SQOPSO respectively.

- NGOpt,

- b-cmafmin2 (the baseline CMA included in the code of BBOB/COCO),

- u-CmaFmin2 and r-CmaFmin2 correspond to the same CMA but with different restart schemes (the same restart as for Scipy methods for rCma and uniform random restarts for rCma, whereas CmaFmin2 uses a proposal function "propose-x0" from the objective in Coco/Bbob),

- and other variants of CMA found in Nevergrad (see Appendix I for more information).

We have 24 cases, corresponding to dimensions 2, 3, 5, 10, 20, 40 and budget/dimension 10, 100, 1000, 10000. NgIoh4 and NgIoh21 both outperformed NGOpt in 19/24 cases, NgIohTuned outperformed NGOpt in 21/24 cases. NgIoh4 outperforms bCmaFmin2 in 14/24 cases. NgIoh21 outperforms bCmaFmin2 in 17/24 cases and is frequently the best overall. NgIohTuned outperforms bCmaFmin2 in 19/24 cases and is frequently the best overall. NgIoh21 and NgIohTuned are frequently very close, for a reason: note that on BBOB/COCO (i.e. artificial benchmarks), they are equivalent. Overall, NgIoh4 and all its variants outperform NGOpt on BBOB/COCO.

Results in low budget cases confirm the excellence of Cobyla already observed in (Dufossé & Atamna, 2022; Raponi et al., 2023).

### H.3.1 Additional results: BBOB with budget $= 10\times$ dimension

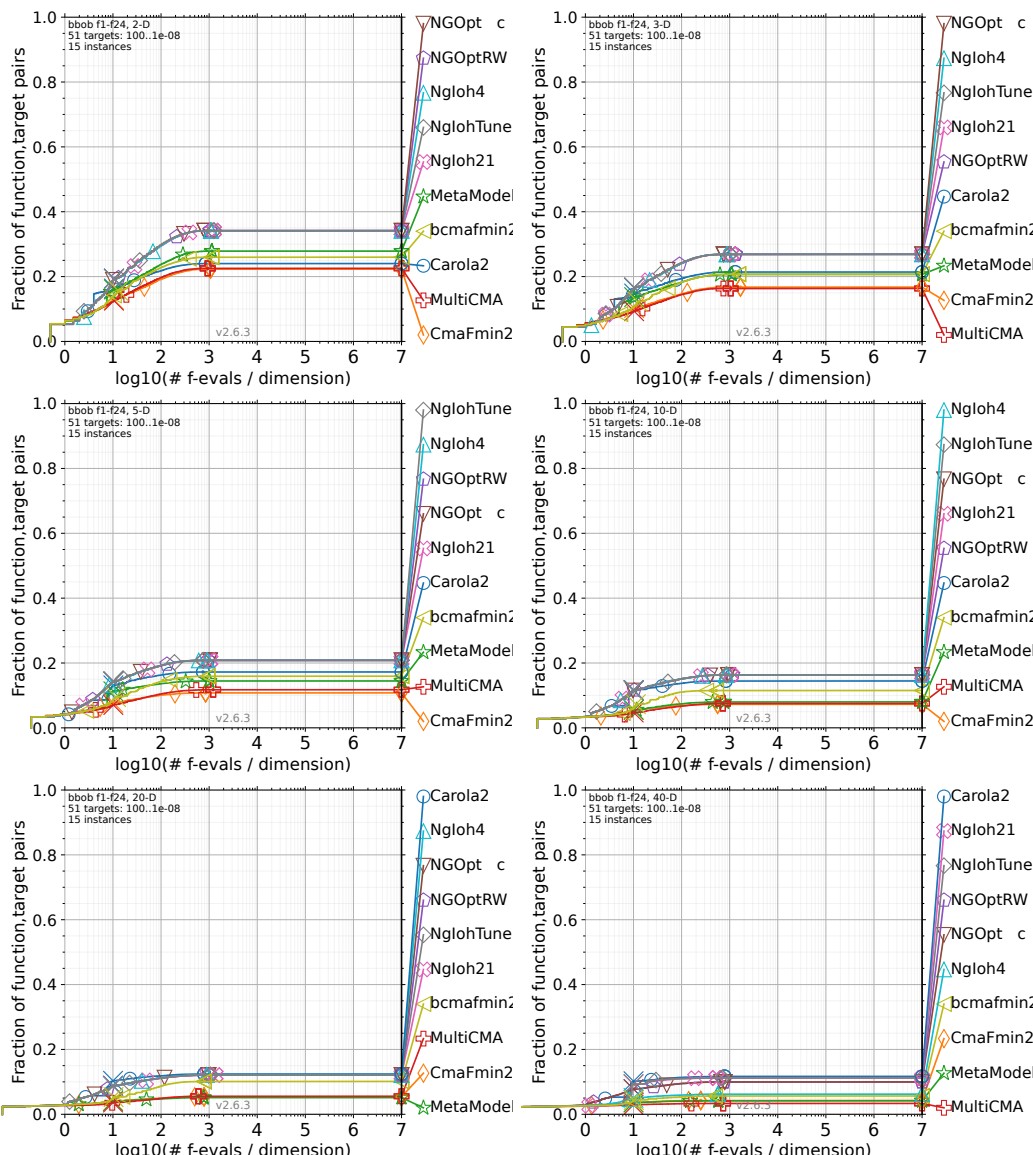

Figure 21: BBOB with budget $10\times$ dimension. The higher the better on these BBOB/COCO figures.

### H.3.2 Additional results: BBOB with budget $= 100\times$ dimension

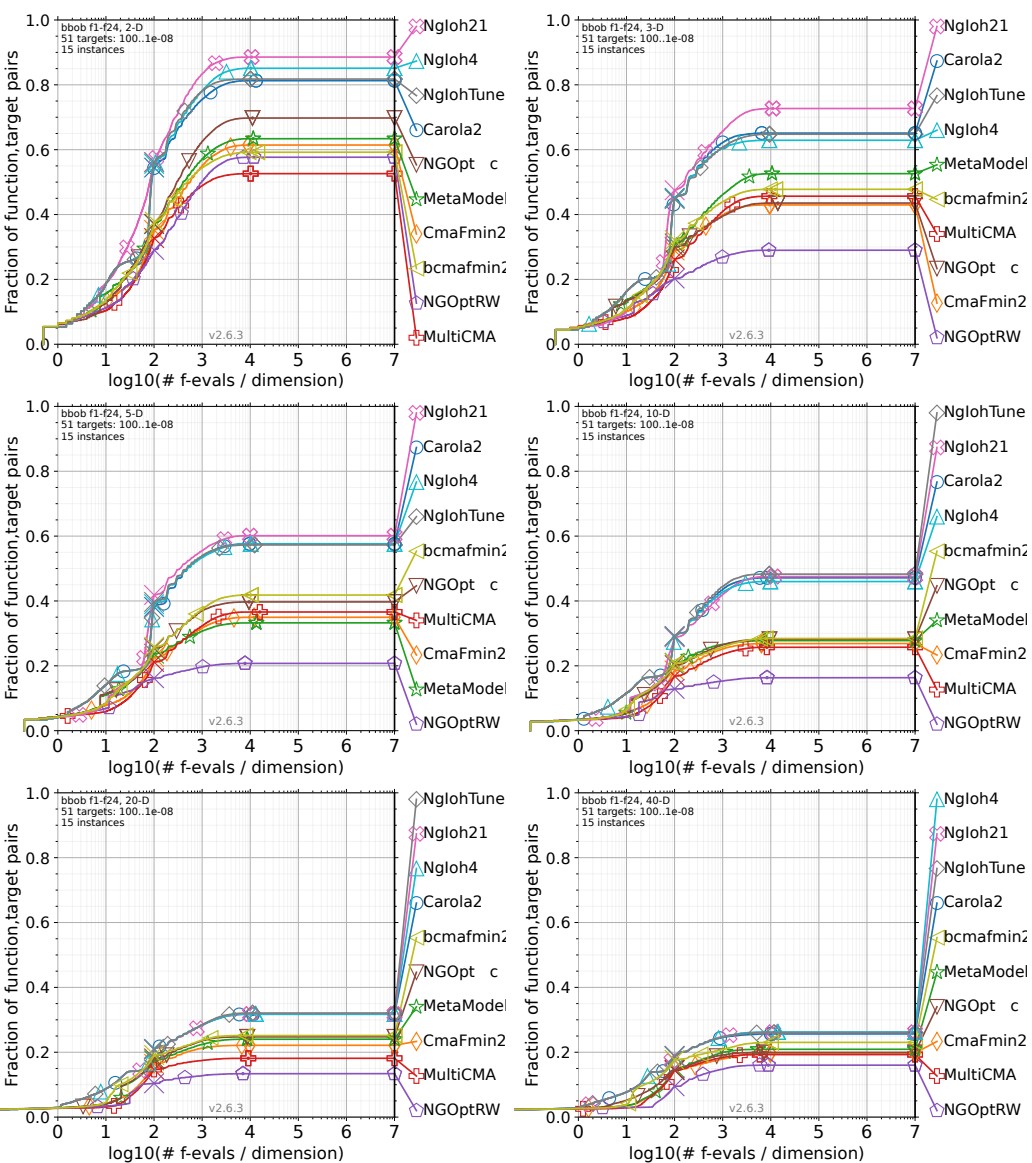

Figure 22: BBOB with budget $100\times$ dimension. The higher the better on these BBOB/COCO figures.

### H.3.3 Additional results: BBOB with budget $= 1000\times$ dimension

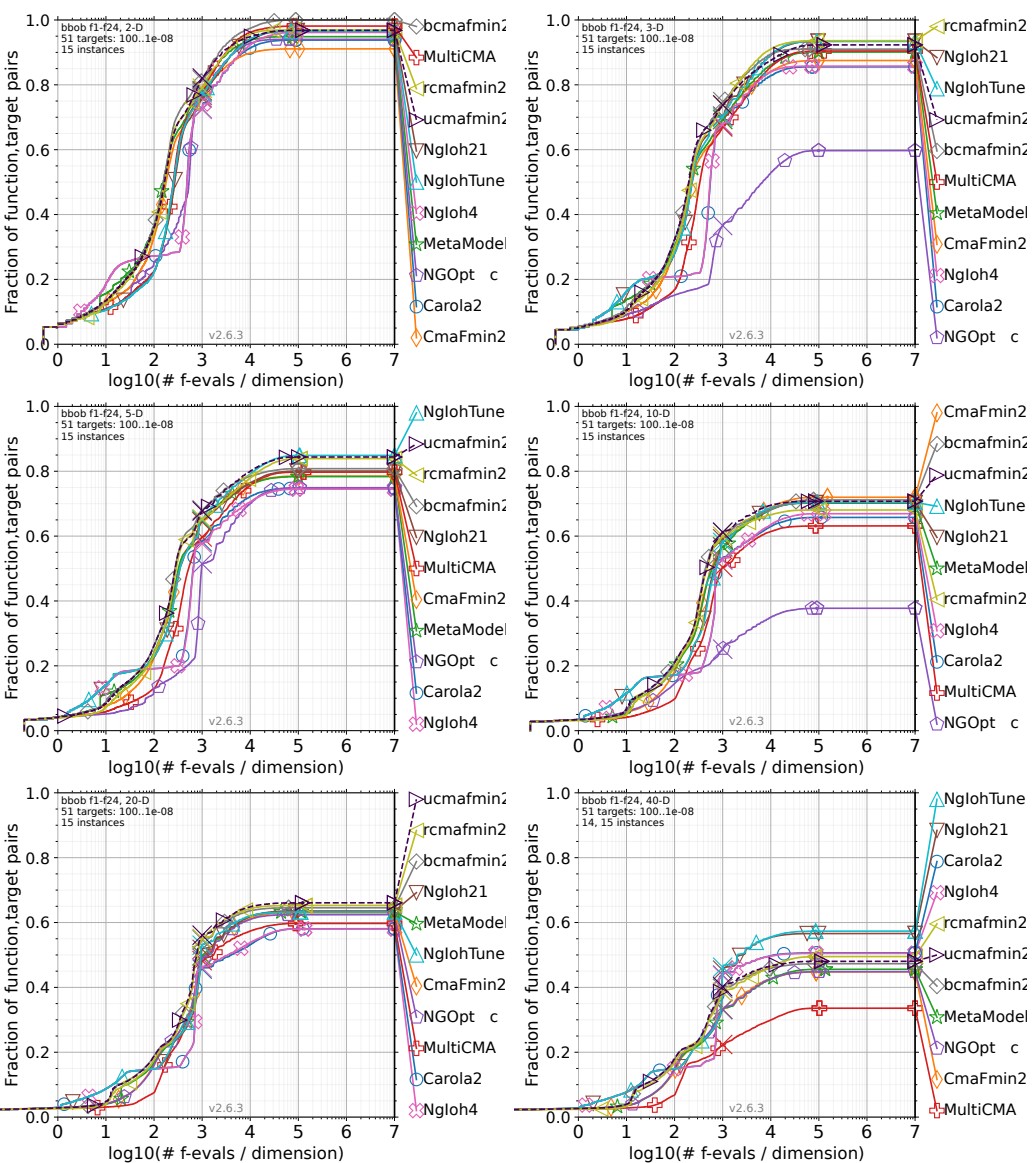

Figure 23: BBOB with budget $1000\times$ dimension. The higher the better on these BBOB/COCO figures.

## H.3.4 Additional results: BBOB with budget $= 10000\times$ dimension

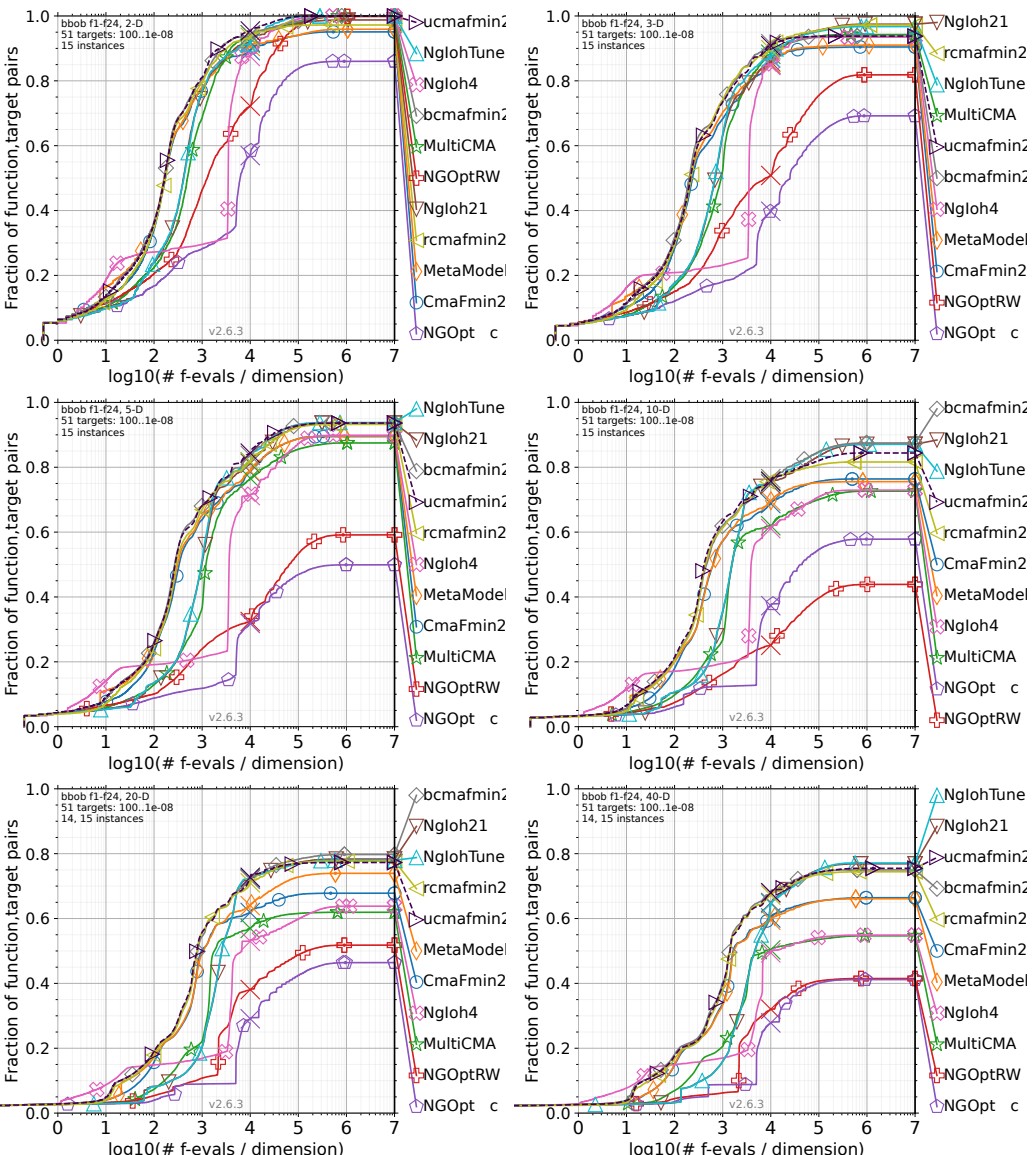

Figure 24: BBOB with budget 10000× dimension. The higher the better on these BBOB/COCO figures.

# I List of methods

All details and implementations of the algorithms discussed here and more are available at Rapin & Teytaud (2024).

- Carola1, Carol2, Carola3 (Section 3): chaining of Cobyla, MetaModel and SQP(Virtanen et al., 2020). Carola3 is the parallel version. Other variants based on a warmup by Cobyla have been tested (Carola4+ in the codebase) without significant improvement.

- Cobyla: Mathematical programming method based on linear interpolation (Powell, 1994).

- DSproba: the direct-search method in Section 2.2. Other variants include DSbase, DSsubspace, DS3p.

- MetaModel: CMA, equipped with a Metamodel. The meta-model is applied periodically, if and only if the quadratic model can effectively learn the loss values. There are variants with a random forest meta-model or a neural-net meta-model.

- ChainMetaModelSQP: a chaining of Meta-Model (most of the run) and SQP (fast local search at the end).

- CMA: Covariance Matrix Adaptation Evolution Strategy (oldCMA is an older variant). DiagonalCMA is a version which is faster thanks to the usage of a covariance matrix. MicroCMA is a variant with an initialization with very small variance. MultiCMA and PolyCMA are bet-and-runs(Weise et al., 2019) of several CMA with different initializations. LPCMA is a CMA with larger population and VLPCMA has even greater population. PymooBIPOP is the BIPOP-CMA as implemented in Pymoo(Blank & Deb, 2020). ECMA is a so-called elitist variant of CMA, and FCMA is the implementation of CMA in (Wolz, 2022).

- DiscreteLenglerOnePlusOne: a discrete optimization method in (Einarsson et al., 2019) for a decreasing schedule of the mutation rate. BSODiscreteOnePlusOne is similar but with a handcrafted schedule instead of a mathematically derived one.

- Other discrete optimization methods include FastGA (Doerr et al., 2017b), Lognormal variants (Kruisselbrink et al., 2011a), Portfolio (based on (Dang & Lehre, 2016)). Variants of these methods with various suffixes or prefixes correspond to modifications or different parametrizations. Adaptive variants from (Doerr et al., 2019; Einarsson et al., 2019; Doerr et al., 2016; 2017a) are also available, with Adaptive in their name.

- DE: differential evolution. GeneticDE, Two-PointsDE and others are defined in Section 2.2. Variants include QNDE, QODE, SPQODE defined in Appendix C. MemeticDE is a chaining of RotatedTwoPointsDE, TwoPointsDE, DE, SQP.

- PSO: Particle Swarm Optimization. Variants include QOPSO and SQOPSO, defined in Appendix C, and SQOPSODCMA which chains SQOPSO and DiagonalCMA.

- HyperOpt (Bergstra et al., 2015): optimization method based on Parzen estimators.

- NGOpt: the classical wizard from Nevergrad. Shiwa is an old variant of NGOpt. Specialized variants exist: NGOptRW: a wizard in Nevergrad, created specifically for real-world cases. NoisyRL (with various suffixes): variants specialized for reinforcement learning with noise. MixDeterministicRL: variant for deterministic RL.

- The BAR algorithms are bet-and-runs combining DE, CMA and others (BAR1, BAR2, BAR3, BAR4): they are detailed in the codebase and are, overall, weaker than wizards.

- NgIoh2, NgIoh4, NgIoh5, NgIoh6: variants of NgIoh, not using information about the type of variables or the type of problems. We present NgIoh4 in Section 3 and details about the other variants, tested for ablation, are available in (Rapin & Teytaud, 2024): NgIoh5 and NgIoh6 differ from NGIoh4 in the discrete case only and therefore equivalent to NgIoh4 in the continuous case. NgIoh2 and Wiz differ more, are usually weaker, and are visible in the codebase.

- NgIohTuned: variant of NgIoh, using information on the type of variables (e.g.,: neuro-control weight) and the type of problem (e.g.: real world or not). NgIoh21 is similar to NgIohTuned, without using special cases for real-world problems.

- OnePlusOne: The simple $(1 + 1)$-evolution strategy with one-fifth update rule.

- SQPCMA: running in parallel several SQP with different initializations and a CMA.

- SQP: Sequential Quadratic Programming (Nocedal & Wright, 2006, Chapter 18)(Virtanen et al., 2020).

- NLOPT: methods in (Johnson, 1994) (the suffix is the method name).

- MetaRecentering: a method for quasi-random sampling combined with a mathematically derived guess of the location of the optimum (Meunier et al., 2021).

- pysot: a tool including Bayesian optimization methods and others (Eriksson et al., 2019).

- Zero: a baseline which always returns the center of the domain. Loss values are ignored.

- TBPSA: Test-Based Population Size Adaptation (Cauwet & Teytaud, 2020b), an algorithm for continuous noisy optimization.

- NaiveTBPSA: an adaptation of TBPSA for the deterministic but highly rugged landscape (Cauwet & Teytaud, 2020a).

- Stupid: a baseline which randomly draws a point in the domain and returns it as a recommendation. Loss values are ignored.

- Many of these methods have counterparts based on combinations with optimism in front of uncertainty as in the bandit literature.

