# OpenReview forum: "NgIohTuned: a Highly Reproducible Black-box Optimization Wizard"
_TMLR — Rejected by TMLR_

### Review · Reviewer_w7wK · 2024-11-06

**Summary Of Contributions:**

The paper "NgIohTuned: a Highly Reproducible Black-box Optimization Wizard" proposes new benchmark problems for black-box optimization. The benchmark problems address an open problem of "scale" in benchmarking. The paper then evaluates a plethora of optimization algorithms on the problem, finding that differential evolution and particle swarm optimization perform among the best.

**Audience:**

Yes

**Broader Impact Concerns:**

No concerns.

**Claims And Evidence:**

No

**Requested Changes:**

# Major
* Describe the experimental setup in the main paper. It is unclear to me how many repetitions (named replica in the paper) were used per method, what the budget was, and how both were chosen.
* Properly define "scale" of a black-box optimization problem. It might also be helpful to give a few examples, or "survey" the existing problems for their scale. Then, demonstrate that the proposed, new problems actually exhibit better "scale".
* Separate background from contributions: For example, the 2nd paragraph under 2.1 ends with the statement that you add a new method. Similarly, in the 4th paragraph, the paper discusses new benchmarks. I do not see how this is background information, and such mixing of novel content and background makes the paper hard to read, and hard to see the contributions.
* Please separate the description of methods from the description of benchmarks. The last paragraph under 2.1 mixes both, but according to its first sentence only discusses "black-box optimization platforms". Such inconsistencies make the paper really hard to read, please check the whole paper for them.
* Please separate the background from discussing your results. In section 2.2 the paper already discusses results. Also, I have the impression that section 2.2 actually discusses novel optimization methods that were added to Nevergrad. This is extremely confusing to me, because new optimization methods are not part of the contributions listed in the introduction. Also, it is not even clear to me how many of the named methods were invented by the authors (by combining existing concepts) and how many are directly taken from literature. I would suggest to not add new optimization algorithms in this paper, or make a table with new optimization algorithms so that it is clear which ones you invented, and which ones are taken from literature.
* The figures are inconsistent. Please use the same colors for the same methods across different plots, especially if you have two plots next to each other (Figure 1).
* Please refrain from using abbreviations as section headings. I can only guess what section 3 is about.
* Please improve Section 3. I think the paper states that someone ran experiments and analyzed them with IOH. However, the paper currently states that observations made in Doerr et al. (2018) were analyzed to obtain the current findings. Also, move background out of section 3, for example, an explanation of what NGOpt does.
* Please improve the writing of the paragraph "Scaling and distribution of optima in continuous domains: multi-scale problem" in Section 4.2. The paragraph mixes zero with origin (and I think it would be good if the paper only used the term origin as it is unambiguous), does not explain important terminology (only references Kumar, 2017), and uses the term "scale" with a different meaning than the one in section 2 (I assume the term scale refers to a parameter of the optimization algorithms here).
* The paper is written for an audience that attended the Dagstuhl seminar "Challenges in benchmarking optimization heuristics". I am not sure if all potential readers know what a Dagstuhl seminar is, so I would suggest that the paper provides more context and that it is written in a more self-contained manner.
* Every method in Section I (list of methods) should either have a reference paper, or a pointer to the section where they were introduced. By this it would be clear if the method was introduced by the paper, or is a method from the literature.

# Minor
* Please check the text carefully for citations used in text and in brackets. Very often, the citations are in brackets, when they are used as nouns, which makes the text hard to read (see, for example, the citations in the first contribution).
* In the sentence "Compared to CMA (Hansen & Ostermeier, 2003), the method focuses on rapidly approximating the appropriate scale and is well-suited for high-dimensional settings." it is unclear what "the method" refers to. Is it CMA or is it PSO & DE from the previous sentence?
* The method discusses SMAC3 and HyperOpt as successful implementations of Bayesian optimization. However, it misses another really strong contender: HEBO. It would be great to include that in the discussion.
* I am wondering how the term "wizard" relates to the wizards that are typically used in computer programs (please check wikipedia for the definition)?
* Please describe how you came up with the rules for NgIoh4. Is there some general concept that can be used to derive such rules, or were these rules found by experts after looking at the data? Also, how were bounds chosen? Were systematic and reproducible principles used, or are these guesses after looking at the data? Currently, this algorithms comes out of the blue and completely lacks motivation.
* What definition of reproducibility does the paper follow? It would be great if this could briefly be mentioned in a footnote (see https://arxiv.org/abs/1802.03311 for a discussion). Depending on your definition, a lot of the examples under 4.1 might not be poor reproducibility, but rather poor experimental setup. You maybe want to also reference a recent ICML paper discussion current problems in machine learning research in a more general setting (https://proceedings.mlr.press/v235/herrmann24b.html).
* The sentence "We propose a code fully available in open access." is ambiguous. Do you propose that researchers open source their code, or do you propose methods for which you provide open source code? In general, please check the full text for such ambiguouties.
* The text continues with "As these URLs are automatically updated", which is factually wrong. The URLs are not updated, the content that is linked is updated.
* The next part is worrisome: "they [the data] might differ thanks to additional work by contributors and re-runs". Does this mean that the results are subject to change, and what we are reviewing right now might not be what is in the final paper? Please also refer to my first bullet point under major request changes.
* The sentence "Ungredda et al. (2022); Dagstuhl participants (2023) showed that cases with budget with more than 100 times the dimension (which is frequent in artificial benchmarks) might be the exception rather than the norm." is ambiguous as it does not specify if this applies to academic or real-world benchmarks.
* I do not understand the sentence "Note that some benchmarks do not have the same functions for the different values of the budget." How can the same benchmark have different functions? Isn't the goal of a benchmark to always provide the same function to optimize?
* Please avoid repetition. A lot of details on the benchmarks in 5.1 were already introduced in 4.2.
* I do not understand the sentence "These results indicate that the results are not deteriorated when compared to NGOpt." What does that mean?
* It is not clear how the algorithms in the plots were chosen, and why the number of algorithms under comparison changes. I would suggest that the paper focuses on one thing, and does not try to introduce new benchmarks and new algorithms, and then conduct a large-scale study on them. I would appreciate if the paper only demonstrated the necessity for the new benchmark problems and used standard optimization algorithms to show how they behave on these new problems.
* Please correct the reference to the Black-box optimization challenge, the correct paper is https://proceedings.mlr.press/v133/turner21a.html
* Please be consistent in abbreviating (or not abbreviating) conferences in the references. Also, be consistent in giving DOIs and ISBNs.
* The reference Haibe-Kains is broken.
* What do you mean by "One of the conclusions from this experiment is how much most Bayesian methods cannot be used for large budgets and dimension spaces larger than 84"? Do the methods in Nevergrad fail, or is this a general problem?
* Is the Crop optimization benchmark (G.2) a new benchmark that is part of the contributions? If yes, I am surprised by the fact that it will only be described in a forthcoming publication.

**Strengths And Weaknesses:**

Strengths:
* New test functions for black-box optimization.
* The paper is about reproducible research, which is an important topic in machine learning.

Weaknesses:
* The paper is incredibly hard to read.
* Besides the two contributions mentioned in the introduction, the paper has a third contribution: it introduces several new algorithms without clearly describing or benchmarking them.
* No contextualization of results. Results are presented, but not discussed wrt what one would expect from the benchmarked methods.

---

### Review · Reviewer_yA7u · 2024-11-15

**Summary Of Contributions:**

The authors study various problems in black box optimization (BBO) - optimization problems where there is no gradient/objective information available outside of evaluation on specific points. They develop a new set of benchmarks that induce optima at multiple scales, as well as remove information from evaluations near 0. They also introduce a new BBO, NgIohTuned, and benchmark it and other algorithms and find that DE, PSO, bet-and-run, and NgIohTuned all perform well across many problems.

**Audience:**

No

**Claims And Evidence:**

No

**Requested Changes:**

The overall structure needs to be changed:

* A better overview of BBO, as well as establishing notation earlier on that can be referenced later
* Better descriptions of the key algorithms
* More details about NgOpt and why it is the basis of the new method
* Figures more readable (e.g. Figure 1 is impossible to parse)
* Overall cleanup of the sentence-by-sentence language to meet scientific writing style

**Strengths And Weaknesses:**

The idea of developing multi-scale benchmark problems seems quite important; this was the most compelling section of the paper. Knowing the scale of good solutions (or indeed, distribution of scales) in real world problems is difficult, especially without gradient information. In addition the zero-penalized benchmark seems well motivated.

However, the overall writing quality is not up to standard for TMLR submissions (let alone accepted papers). There was not enough structure to the early part of the paper; it seemed to assume not only expertise in BBO at large but also expertise in minute details of every algorithm. The paper requires a more structured introduction to the topic and algorithms, and more motivation for the key contributions. It was very unclear where different sections were going, or what I was to take away from them. Also, without any previously established notation it was difficult to understand what some of the described algorithms were doing.

There were also many many sentence-by-sentence issues in the writing, e.g.:

>In discrete optimization, the good
old 1/d mutation consists in randomly mutating each variable with probability 1/d in dimension d. Typically,
a single variable is mutated; it rarely includes more than two variables. Some algorithms, in particular after
the good results described in (Dang & Lehre, 2016), use a fixed random distribution of mutation rates.

Lines like this were both too causally written for scientific writing style, and were also uninformative (e.g. what were the good results in Dang and Lehre?).

The figures were also very very difficult to parse. For example, both the lines and the labels in Figure 4 were too difficult to understand.

In addition, the new proposed BBO, NgIoh4, seems like a trivial extension of previous work (applying one of two previously known optimizers, with a very simple choice function which doesn't depend too much on problem details). I am doubtful this will generate interest from the TMLR readership.

Overall, I recommend that the authors do a complete rewrite of the paper with some of these high-level points in mind. I believe that such changes are beyond the scope of the normal TMLR review cycle.

---

### Review · Reviewer_5V9d · 2024-11-27

**Summary Of Contributions:**

The paper introduces a new black-box benchmarking framework containing real-world and synthetic benchmarks to improve reproducibility. It also presents a study comparing existing state-of-the-art methods from the literature and provides a new optimization wizard that selects between different optimizers based on the input dimension.

**Audience:**

Yes

**Broader Impact Concerns:**

I don't see any ethical concerns of the paper

**Claims And Evidence:**

No

**Requested Changes:**

- The introduction should motivate the problem and provide a brief overview of the problem setting.
- Mathematically define important concepts that are used throughout the paper, such as black-box optimization and scale.
- Increase the font size of the plots and consider showing a subset of the methods to improve clarity.
- Place plots side-by-side where appropriate to save space.
- Add a clear overview of all benchmarks and methods (e.g., in the form of a table) to the main text.

**Strengths And Weaknesses:**

## Strengths:

- Classical black-box optimization is an important problem with high practical relevance and is arguably somewhat underrepresented in the ML community.

## Weaknesses:

-  Overall, I found the paper to be rather poorly written and very hard to follow:
        - The introduction jumps directly into the contributions without properly motivating and introducing the problem setting.
        - The introduction claims that black-box optimization benchmarks are formalized in Section 4.1 (which is far too late), but Section 4.1 focuses entirely on reproducibility.
        - The paper contains a significant amount of jargon (e.g., optimization wizard, bet-and-run) and undefined acronyms (e.g., CMA, SMAC3, COCO), making it difficult to understand for readers not deeply familiar with the subject.
        - The plots are cluttered, making them hard to interpret.

 - The paper does not reference the rich literature on benchmarking in AutoML, such as hyperparameter optimization and neural architecture search, which also represent black-box optimization benchmarks and appear to resemble more real-world problems. For example:

NAS-Bench-201: Extending the Scope of Reproducible Neural Architecture Search
Xuanyi Dong, Yi Yang

YAHPO Gym - An Efficient Multi-Objective Multi-Fidelity Benchmark for Hyperparameter Optimization
Florian Pfisterer, Lennart Schneider, Julia Moosbauer, Martin Binder, Bernd Bischl
Proceedings of the First International Conference on Automated Machine Learning, PMLR 188:3/1-39, 2022.

- Algorithm 2, which is one of the contributions of the paper, appears to consist only of a simple if/else statement.

- Section 4.2: I am not convinced by the argument that "dependency upon a specific version of a given package is detrimental to reproducibility." To me, the whole point of versioning in software engineering is to enhance reproducibility.

---

### Decision · Action_Editor_7wif · 2025-01-03

**Recommendation:** Reject

**Comment:**

The paper is not meeting TMLR presentation and technical standards. The manuscript needs significant revisions (eg, extensive use of jargon, unspecified concepts, etc.) and is missing references. Reviewers indicated that claims were not supported. Authors did not address the comments raised by the reviewers.

**Audience:**

Black-box optimisation is of general interest to the ML/AI community and publications in this area are relevant to the TMLR community. Unfortunately, the reviewers found that the findings in the paper were insufficiently supported and not in a state they can be shared with the community.

**Claims And Evidence:**

The authors consider black-box optimisation problems and develop a new set of benchmarks. They also propose a new method for black-box optimisation, comparing it with a number of baselines from the literature. All reviewers raised concerns regarding the results presented in the paper and found that the claims were not supported. Reviewers also indicated that the manuscript's presentation was not up to the standards of TMLR.